# Residual Connections Relay Generalization but Not Memorization in Transformers

## Abstract

Residual connections are one of the main components in transformers, helping stabilize training and improve optimization, yet it remains unclear how they influence memorization, a behavior that transformers are known to exhibit, especially in overparameterized regimes. Therefore, in this work, we investigate the impact of residual connections on memorization in transformers. Our analysis shows that residual connections do not influence memorization; instead, their removal primarily impairs learning, which is a *novel* finding. Furthermore, we find that residual connections in early layers are significantly more important for performance than those in later layers. To explain these findings, we perform a gradient flow and output margin analysis, demonstrating how residual connections support learning dynamics without propagating memorization.

## 1 Introduction

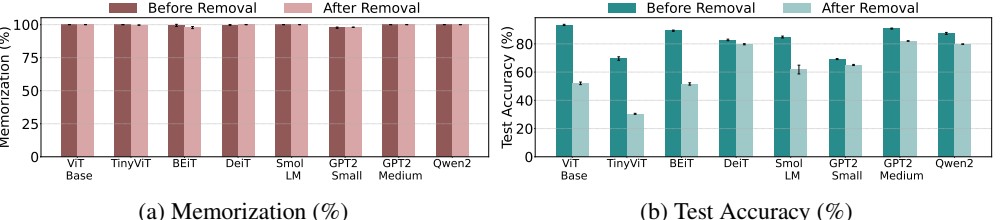

|           |
|-----------|
| (a) Memorization (%) |  (b) Test Accuracy (%) |

Figure 1: **Residual Connections have no impact on memorization.** (a) Residual connections do not relay memorization, as indicated by nearly identical 100% memorization before and after removal (red-bars) (b) Removal of residual connections impairs learning (green-bars), leading to significant drops in generalization performance. (Removing residual connections from layer 1)

Residual connections, first introduced by He et al. (2016) for deep convolutional networks (ResNets), have become a key component of modern deep neural networks because they stabilize the training of very deep models by mitigating vanishing gradients and facilitating smoother gradient flow. By introducing identity mappings to relay the previous input to the next layer along with the main flow, they preserve information across layers and stabilize optimization. Due to their effectiveness, residual connections have also been adopted in the transformer architecture, which enables direct propagation of input information past the attention and feed-forward sublayers.

Although transformers have shown remarkable success in learning complex patterns, they are also prone to memorizing data, a phenomenon often referred to as **label memorization** (Feldman, 2020; Feldman & Zhang, 2020), which acts as a hindrance to the model's generalization ability because it focuses on merely fitting training labels rather than learning meaningful, generalizable patterns. Although there have been some studies considering memorization in transformer architectures (Haviv et al., 2022; Stoehr et al.), residual connections' impact on memorization has never been studied in transformers. This is particularly important because residual connections propagate information, directly from one layer to the next, bypassing intermediate transformations. Such information may contain a mixture of generalizable patterns and memorized signals. Hence, it motivates us to ask the following question:

*"Whether residual connections carry over memorization or not?"*

To answer the question, in our paper, we systematically investigate the influence of residual connections on memorization and learning, respectively, in transformer models. We identify that, surprisingly, residual connections have **no impact on memorization** and only influence learning, and explain the phenomenon through the lens of gradients. We then localize learning across different regions of the network architecture—early, middle, and later residual connections, where we find out that **early residuals are critical for learning**, supported by their higher gradient norms and significant drop in output margins and test accuracy when removed. In summary, the core findings of our paper are as follows:

- **Residual connections have *no* impact on memorization:** We identify that residual connections in transformers do not contribute to memorization and rather only impact learning.
- **Gradients explain why:** We explain why residual connections have no impact on memorization but primarily influence learning by investigating gradients across layers.
- **Early Residual connections are critical for learning:** Our analysis through gradients, standard deviation of residual connections, and output margin reveals that early layers residual connections are the most significant, where there removal destabilizes learning.

## 2 RELATED WORKS

**Memorization & Learning:** Transformers are highly effective in capturing broad, generalizable patterns from training data (Arpit et al., 2017; Shah et al., 2020; Zhou & Wu, 2023), yet they also tend to memorize atypical, noisy, and/or too complex examples (Stephenson et al., 2021; Baldock et al., 2021; Agarwal et al., 2022; Maini et al., 2023), a phenomenon known as *label memorization* (Feldman & Zhang, 2020; Feldman, 2020). Several metrics have been proposed to identify memorized samples, including prediction depth (Baldock et al., 2021), the EL2N score (Paul et al., 2021), and input curvature (Jiang et al., 2020; Ravikumar et al., 2024; Garg et al., 2023). Beyond identification, recent works have sought to localize factual recall knowledge within feedforward and self-attention layers in the transformer architecture, (Dai et al., 2021; Haviv et al., 2023; Geva et al., 2023; Stoehr et al., 2024; Menta et al., 2025), and developing mitigation techniques. Additionally, recent studies have also shown how deeper layers in transformers have limited effect on their learning ability (Yin et al., 2023; Lad et al., 2024; Men et al., 2024; Li et al., 2024; Sun et al., 2025). Despite these insights, no prior work has explored to verify *whether residual connections influence memorization or not.*

**Residual Connections:** Training deep neural networks has historically been hindered by the problem of vanishing gradients (Bengio et al., 1994; Pascanu et al., 2013), where gradients diminish as they propagate through many layers. To address this, residual connections were introduced by He et al. (2016), providing a shortcut that adds the input of a layer (block) to its output, a special case of highway networks (Srivastava et al., 2015) This simple mechanism stabilized gradient flow and enabled the successful training of very deep models (Huang et al., 2020). Several theoretical works have further demonstrated their benefits: Hardt & Ma (2016) established convergence guarantees in deep linear residual networks; Liu et al. (2019) showed that residuals help avoid spurious local optima in convolutional settings; Scholkemper et al. (2024) found that they alleviate oversmoothing in graph neural networks; Veit et al. (2016), showed that gradients flowing through the residual connections have the most impact on training, Hence, due to their effectiveness, residual connections have also been adopted in transformer architectures (Vaswani et al., 2017; Xiong et al., 2020; Dosovitskiy et al., 2020). They are critical for stable optimization and have recently been shown to prevent rank collapse (Dong et al., 2021). Yet, despite growing interest, their *influence on memorization in transformers remains unexplored*—the gap this work aims to close.

## 3 PRELIMNARIES

In the transformer architecture (Vaswani et al., 2017), residual connections, also known as skip connections or identity mappings, are adopted from the original ResNet architecture (He et al., 2016). They operate by directly adding the input of a transformation $\mathcal{F}(\cdot)$ to its output. Formally, given an input $x$ and the transformation $\mathcal{F}(\cdot)$, the output of the residual block is given by,

$$\text{Residual output} = x + \mathcal{F}(x) \tag{1}$$

Each layer in the transformer architecture consists of two residual connections: one surrounding the multi-head self-attention (MHSA) sub-layer and another surrounding the feedforward network

(FFN) sub-layer. These residual pathways enable the direct flow of input information across layers, supporting both gradient propagation and information preservation. In this paper, we investigate the pre-layer normalization architecture (Xiong et al., 2020), due to its superior training stability and its widespread adoption in modern large-scale models such as GPT, Qwen, and LLaMA. The formal description of each residual block for the $i^{\text{th}}$ transformer layer is as follows:

$$\text{MHSA Residual output:} \quad \tilde{x}_i = x_i + \text{MHSA}(\text{LN}(x_i))$$
$$\text{FFN Residual output:} \quad y_i = \tilde{x}_i + \text{FFN}(\text{LN}(\tilde{x}_i)) \tag{2}$$

where $x_i$ and $\tilde{x}_i$ are the inputs (also the residual connections) of the MHSA and FFN residual blocks, respectively, $y_i$ is the output of $i^{\text{th}}$ transformer layer, and LN is the layer-normalization operation.

### 3.1 FORMALIZING LABEL MEMORIZATION (LM) AND LEARNING

Deep neural networks, including transformers, tend to learn rich and meaningful representations between features and labels, which are then generalized to unseen test data, a phenomenon commonly known as ***generalization/learning***. Despite this capability of learning rich patterns, these models also strive to minimize the error based on the empirically seen data samples during training, which is based on Empirical Risk Minimization (ERM) (Vapnik, 1998). Hence, they exhibit a tendency to memorize specific training examples without capturing underlying patterns that can be generalized to the test set, a behavior known as ***label memorization (LM)*** (Feldman, 2020; Feldman & Zhang, 2020), which ultimately leads to overfitting. This behavior occurs due to several factors, such as the presence of noisy labels and/or samples that are overly complex or ambiguous (Baldock et al., 2021), which hinder the model's ability to extract meaningful patterns.

In this study, we investigate memorization by introducing ***noisy labels*** (Maini et al., 2023; Feldman, 2020) in the training set. Specifically, we reassign the labels of a subset of training samples to an incorrect, random label which differs from the true class labels. To ensure that the model memorizes the noisy labels, we train the model until it achieves 100% training accuracy. We conduct experiments under multiple noise ratios: 1%, 5%, 10%, and 20%. We validate our claim also on generative language modeling tasks showing that our results are consistent across various types of tasks.

### 3.2 METRICS TO MEASURE MEMORIZATION AND LEARNING

To study the impact of residual connections on memorization and learning in transformers, we focus on two key metrics, **Test Accuracy (%)** and **Memorization (%)**, as described below.

**Test Accuracy (%)** indicates the model's ability to generalize to unseen data by evaluating its predictions on the held-out test set. It is formally computed as the ratio (%) of the number of correctly predicted samples over the total samples in the test set.

**Memorization (%)** quantifies the extent to which the model fits to mislabeled or corrupted training examples, instead of capturing meaningful patterns. A higher score implies effective memorization of noisy data, rather than true learning. The metric is formally defined as the ratio (%) of the number of correctly predicted samples over the total number of noisy samples.

### 3.3 DATASETS AND MODELS USED

We extensively verify all of our claims on both vision and language modalities as follows:

**Datasets:** Emotions (Saravia et al., 2018), 20NewsGroup (Lang, 1995), TweetTopic (Antypas et al., 2022), CIFAR10 (Krizhevsky et al., 2009), CIFAR100 (Krizhevsky et al., 2009), Places365Mini (Zhou et al., 2017), and UTK-Face (Zhang et al., 2017).

**Models:** GPT2-Small (Radford et al., 2019), GPT2-Medium (Radford et al., 2019), Smol-LM (Allal et al., 2025), Qwen2 (Team, 2024) ,ViT-Base (Dosovitskiy et al., 2020), TinyViT (Wu et al., 2022), BEiT (Bao et al., 2021), and DeiT (Touvron et al., 2021).

## 4 RESIDUAL CONNECTIONS HAVE NO IMPACT ON MEMORIZATION BUT ONLY INFLUENCE LEARNING

To evaluate the role of residual connections in transformer models, we conduct a comparative study by training two model variants: one with residual pathways preserved, and another with these connections explicitly removed (both residual connections removed per layer). We assess their behavior

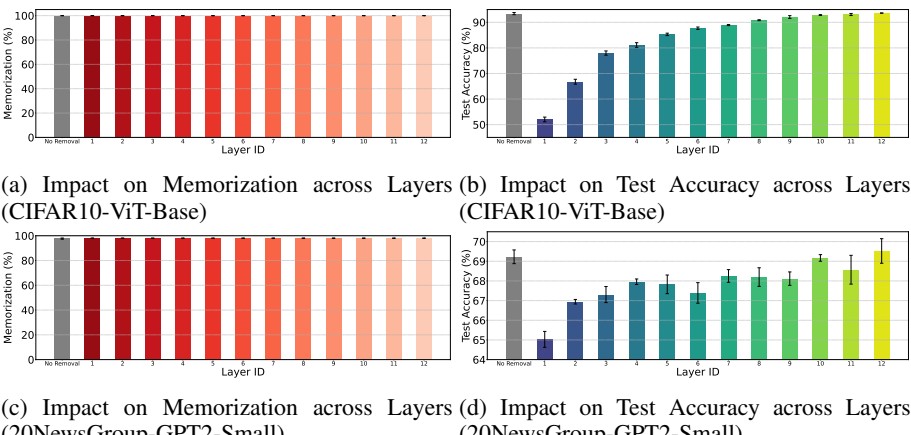

(a) Impact on Memorization across Layers (CIFAR10-ViT-Base)

(b) Impact on Test Accuracy across Layers (CIFAR10-ViT-Base)

(c) Impact on Memorization across Layers (20NewsGroup-GPT2-Small)

(d) Impact on Test Accuracy across Layers (20NewsGroup-GPT2-Small)

Figure 2: **Residual connections do not influence memorization, but early residuals are critical for learning.** (a) shows that residual connections across all layers have no impact on memorization, while (b) highlights that early layers residuals significantly influence test accuracy, indicating their importance for learning, than later ones. The other models' results are provided in Appendix E.1.

using two key metrics—**Test Accuracy (%)** to gauge generalization/learning, and **Memorization (%)** to quantify the extent of label memorization.

### 4.1 MEMORIZATION IS *Not* IMPACTED

From Fig. 1, we observe a surprising result: *residual connections have no discernible impact on memorization* - where memorization consistently remains at 100%, across all models. Specifically, the removal of residual connections does not mitigate the model's tendency to memorize noisy label samples. This trend is further reinforced when examined across different layers. As illustrated in Figs. 2a & 2c for ViT-Base, and GPT2-Small, respectively, the removal of residual connections from any layer does not alleviate the memorization of noisy labels. Consistent patterns are also observed across various transformer architectures, including GPT2-Medium, Smol-LM, Qwen2, TinyViT, BEiT, and DeiT, as presented in Appendix E.1.

### 4.2 LEARNING IS IMPACTED

In stark contrast to memorization, we find that *residual connections primarily facilitate learning*. As shown in Fig. 1, the removal of residual connections significantly degrades the model's ability to learn generalizable patterns, reflected in a substantial drop in test accuracy. Furthermore, in Figs. 2b & 2d for ViT-Base and GPT2-Small, respectively, we observe that models with residual connections removed fail to generalize effectively, with learning severely affected, especially when early layers' residuals are removed. Similar trends are also seen across other transformer models, including GPT2-Medium, Smol-LM, Qwen2, TinyViT, BEiT, and DeiT, as reported in Appendix E.1.

### 4.3 CONSISTENCY ACROSS VARYING LABEL NOISE RATIOS

To further support the robustness of our results, we also analyze higher label-noise ratios of 5%, 10%, and 20%. The result of 20% noise ratio for Smol-LM is shown in Figs. 3a & 3b, confirming the claim that *residual connections relay generalization but not memorization in transformers*. We further validated these claims for other noise ratios as well, 5% and 10%, and DeiT vision model, as provided in Appendix E.2.

### 4.4 CONSISTENT OBSERVATIONS IN GENERATIVE TASKS

To reinforce our claims beyond classification settings, we also evaluate a generative language-modeling tasks. We convert the TweetTopic dataset into a generation task by appending a natural-language prompt of the form: *The topic is about <label>* to each sequence. To examine memorization, we randomly replace the <label> in a subset of sequences with an incorrect one. To evaluate whether memorization of noisy sequences persists or not, we follow the extractable memorization

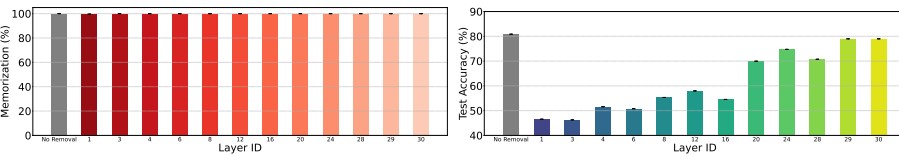

(a) Impact on Memorization across Layers (TweetTopic-Smol-LM & 20% Label Noise) (b) Impact on Test Accuracy across Layers (TweetTopic-Smol-LM & 20% Label Noise)

Figure 3: **Consistent results across higher noise ratios.** Residual connections do not influence memorization but only relay generalization even for higher noise ratio (20%). Consistent results are provided for multiple ratios (1%, 5%, & 10%) and models in Appendix E.2.

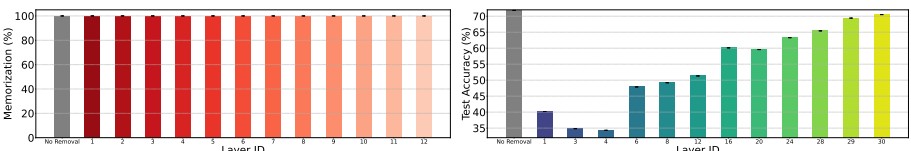

(a) Impact on Memorization across Layers (TweetTopic-Smol-LM & Generative Task) (b) Impact on Test Accuracy across Layers (TweetTopic-Smol-LM & Generative Task)

Figure 4: **Results in generative tasks.** Aligning to classification tasks, residual connections do not propagate memorization even for generative language modeling tasks, but only relay generalization. Additional experiments for GPT2-Small are provided in Appendix E.3.

setting proposed in Carlini et al. (2022). Based on this, we prompt the model with the exact same noisy sequence and ask what topic it is associated with. If the model still outputs the noisy label, then that means memorization still persists even after residual connections removal. We conduct experiments with two models, GPT2-Small and Smol-LM. The results for Smol-LM (Figs. 4a & 4b) show that residual connections do not propagate memorization; instead, they primarily relay generalization. This demonstrates that our findings hold not only for classification tasks but also for generative modeling. Additional consistent results corresponding to GPT2-Small results are provided in Appendix E.3.

### 4.5 GRADIENTS EXPLAIN WHY RESIDUAL CONNECTIONS DO NOT IMPACT MEMORIZATION

To understand why residual connections do not impact memorization, we look into the gradient of the loss function $\mathcal{L}$ with respect to the residual input $x$ (i.e., the input to the residual block), denoted as $g_x = \nabla_x \mathcal{L} = \frac{\partial \mathcal{L}}{\partial x}$, and measure its $\ell_2$ norm, $\|g_x\|_2$ (similarly we compute for the other residual input, $\tilde{x}$, denoted as $\|g_{\tilde{x}}\|_2$). This gradient norm quantifies the sensitivity of a residual connection to either memorization or learning. To assess the impact on learning, we compute $\|g_x\|_2$ (and $\|g_{\tilde{x}}\|_2$) for each test sample and report the average across the test set—referred to as the **learning gradient norm**, $\|g_x^{\text{learn}}\|_2$ (and $\|g_{\tilde{x}}^{\text{learn}}\|_2$). For memorization, we compute $\|g_x\|_2$ over samples with noisy labels and average them to obtain the **memorization gradient norm**, $\|g_x^{\text{mem}}\|_2$ (and $\|g_{\tilde{x}}^{\text{mem}}\|_2$). A high gradient norm at a given layer indicates that its residual connection plays a significant role in learning or memorization, while a low value suggests insignificant influence. As shown in Figs. 5a & 5b, the learning gradient norm, $\|g^{\text{learn}}\|_2$ (where $\|g^{\text{learn}}\|_2 = (\|g_x^{\text{learn}}\|_2 + \|g_{\tilde{x}}^{\text{learn}}\|_2)/2$), is significantly and consistently larger than the memorization gradient norm, $\|g^{\text{mem}}\|_2$ (where $\|g^{\text{mem}}\|_2 = (\|g_x^{\text{mem}}\|_2 + \|g_{\tilde{x}}^{\text{mem}}\|_2)/2$), across all layers. This observation suggests that residual connections primarily aid in the propagation of gradients contributing to generalization, rather than memorization.

Consequently, removing residual connections is expected to impair learning performance while having minimal effect on memorization, as previously observed in Figs. 2b, 2a & Figs. 2d, 2c, where residual removal leads to a notable drop in test accuracy (learning), but leaves memorization accuracy virtually unchanged. These trends hold consistently across a range of architectures, including GPT2-Small, Smol-LM, Qwen2, ViT-Base, BEiT, and DeiT, as further detailed in Appendix E.6. Together, these findings reinforce the notion that residual connections only relay generalization but skip memorization.

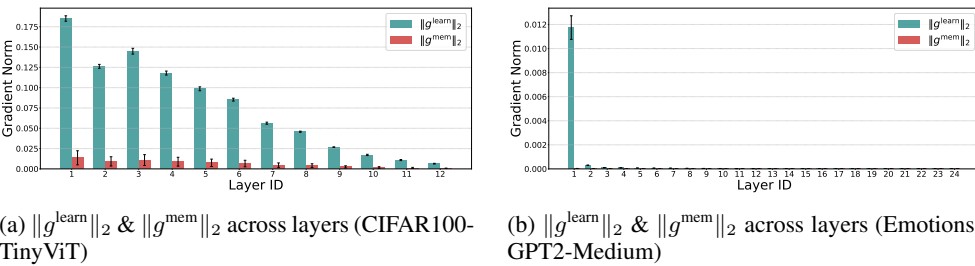

(a) $\|g^{\text{learn}}\|_2$ & $\|g^{\text{mem}}\|_2$ across layers (CIFAR100-TinyViT)

(b) $\|g^{\text{learn}}\|_2$ & $\|g^{\text{mem}}\|_2$ across layers (Emotions-GPT2-Medium)

Figure 5: **Memorization gradient norms are consistently smaller than learning gradient norms, with early residuals exhibiting the highest activity.** $\|g^{\text{mem}}\|_2$ remains significantly lower than $\|g^{\text{learn}}\|_2$ across all layers, explaining why residuals do not influence memorization. The learning gradients peak in early layers, underscoring their critical role in learning.

| Metric | Mean | Std |
|---|---|---|
| $\sigma_{x_j}$ | 2.326 | 2.238 |
| $\sigma_{\tilde{x}_j}$ | 2.304 | 2.253 |
| $\sigma_{W_{j,O}}$ | 0.057 | 0.004 |
| $\sigma_{W_{j,1}}$ | 0.059 | 0.002 |
| $\sigma_{W_{j,2}}$ | 0.058 | 0.009 |

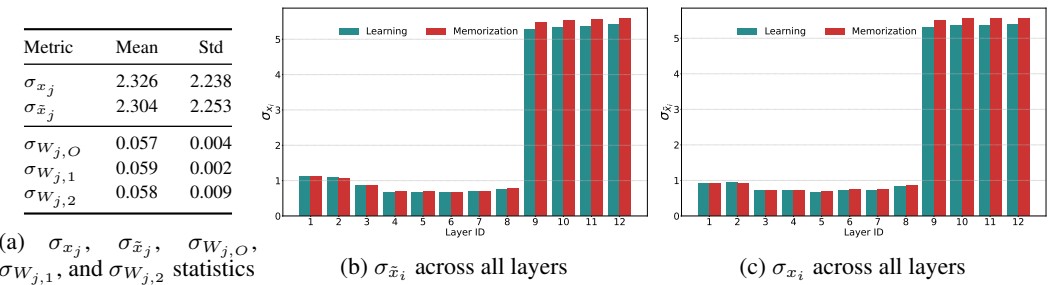

(a) $\sigma_{x_j}$, $\sigma_{\tilde{x}_j}$, $\sigma_{W_{j,O}}$, $\sigma_{W_{j,1}}$, and $\sigma_{W_{j,2}}$ statistics

(b) $\sigma_{\tilde{x}_i}$ across all layers

(c) $\sigma_{x_i}$ across all layers

Figure 6: **Residual block activations exhibit a high variation across layers but remain consistent between learning and memorization.** The standard deviations of residual connections ($\sigma_{x_j}$, $\sigma_{\tilde{x}_j}$) vary substantially across layers, in contrast to the relatively stable statistics of model parameters ($\sigma_{W_{j,O}}$, $\sigma_{W_{j,1}}$, $\sigma_{W_{j,2}}$). Importantly, $\sigma_{x_j}$ & $\sigma_{\tilde{x}_j}$ statistics are nearly identical for learning and memorization samples. (CIFAR100-TinyViT).

Even though gradient norms provide a useful explanation that residual connections do not impact memorization, we still do not know *why the memorization gradient norm is consistently smaller.* Hence, in Sec. 4.6, we investigate this issue in detail, aiming to uncover the factors in structure and optimization process that drive this discrepancy.

### 4.6 WHY ARE THE GRADIENT NORMS SMALLER FOR MEMORIZATION THAN LEARNING?

In Figs. 5a & 5b, we observed that $\|g^{\text{learn}}\|_2$ is significantly and consistently larger than the memorization gradient norm, $\|g^{\text{mem}}\|_2$ across all layers. Going further, in this section, we investigate why memorization samples tend to exhibit lower gradient norms than learning samples. To better understand the reasons behind this consistent disparity, we derive an upper bound on the gradient norm with respect to the residual block input in Theorem 1. The upper bound provides theoretical intuition by breaking the gradient into interpretable components: (i) the prediction error, (ii) residual connections, and (iii) model parameter statistics, while shedding light on why memorization gradient norms are smaller than learning gradient norms across the network.

**Theorem 1 (Upper Bound of Gradient Norm).** *Let $x_i$ be the input to the $i^{th}$ layer's first residual block. Then, the gradient norm satisfies:*

$$\|g_{x_i}\|_2 = \left\|\frac{\partial \mathcal{L}}{\partial x_i}\right\|_2 \leq \underbrace{\|\hat{y} - y\|_2}_{error} \cdot \sigma_{out}\left(\sqrt{d_{out}} + \sqrt{d_1}\right)$$

$$\cdot \left[\prod_{j=i}^{N}\left\{\left(1 + \frac{\sigma_{W_{1,j}}\,\sigma_{W_{2,j}}}{\sigma_{\tilde{x}_j}}\,C_{ffn}\right) \cdot \left(1 + \frac{\sigma_{W_{O,j}}}{\sigma_{x_j}}\,C_{attn}^j\right)\right\}\right] \tag{3}$$

*where $y$ is the ground truth one-hot encoded vector, $\hat{y}$ is the predicted softmax probability vector, $\sigma_{x_j}$, $\sigma_{\tilde{x}_j}$ are the standard deviations of the residual stream inputs $x_j$ and $\tilde{x}_j$, respectively, $\sigma_{W_{1,j}}, \sigma_{W_{2,j}}$ are the standard deviations of the FFN weight matrices $W_{1,j}$, $W_{2,j}$, respectively, $\sigma_{W_{O,j}}$ is the standard deviation of the MHSA output projection matrix, $C_{ffn} = \left(\sqrt{d_1} + \sqrt{d_2}\right)^2$, $d_1$ & $d_2$ are intermediate hidden sizes, $C_{attn}^j = 2d_1 \cdot \left\| J_Z^j \right\|_2$, $\sigma_{out}$ is the standard deviation of classification head weight matrix $W_{out}$, and $d_{out}$ is output size of classification head.*

A formal proof of Theorem 1, along with the expression for $\|g_{\tilde{x}_i}\|_2$, is provided in Appendix A.

Theorem 1 provides an upper bound on the gradient norm while expressing how it depends on - **prediction error**, **residual connections**, and **model parameters** statistics. Specifically, it shows that the upper bound of the gradient norm for any layer $i$ is dependent on (i) prediction error $\|\hat{y} - y\|_2$, (ii) standard deviation of residual connections $x_j$, $\tilde{x}_j$, and (iii) standard deviation of weight matrices $W_{1,j}$, $W_{2,j}$, $W_{O,j}$, and $W_{out}$. This becomes especially useful when comparing gradient behaviors across learning and memorization regimes as discussed below.

As stated previously, to compute the learning gradient norm $\|g^{\text{learn}}\|_2$, we use clean samples (correctly labeled) from the test set, while the memorization gradient $\|g^{\text{mem}}\|_2$ is calculated using noisy labeled training samples. Now, suppose we select one sample from each of these sets, $(x^c, y^c)$ and $(x^{\text{NL}}, y^{\text{NL}})$, where both $x^c$ and $x^{\text{NL}}$ belong to the same true (semantic) class $y^c$, but $x^{\text{NL}}$ is mislabeled as $y^{\text{NL}}$ $(\neq y^c)$. Since both inputs correspond to the same semantic class, the residual connections $x_j^c$ and $x_j^{\text{NL}}$ (and similarly the second residual connection $\tilde{x}_j^c$ and $\tilde{x}_j^{\text{NL}}$) are expected to be similar, i.e.,

$$x^c \approx x^{\text{NL}} \implies x_j^c \approx x_j^{\text{NL}} \implies \sigma_{x_j^c} \approx \sigma_{x_j^{\text{NL}}}; \quad x^c \approx x^{\text{NL}} \implies \tilde{x}_j^c \approx \tilde{x}_j^{\text{NL}} \implies \sigma_{\tilde{x}_j^c} \approx \sigma_{\tilde{x}_j^{\text{NL}}} \quad (4)$$

We also empirically verify this approximation in Figs. 6b & 6c, which show a strong similarity of $\sigma_{x_i}$ and $\sigma_{\tilde{x}_i}$ for both memorization and learning cases, across all layers. This trend also holds across various architectures, including GPT2-Small, GPT2-Medium, Smol-LM, Qwen2, ViT-Base, and BEiT, DeiT, as shown in Appendix E.7.

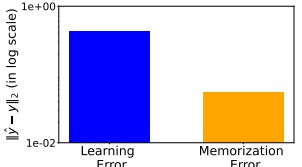

Figure 7: Memorization and Learning error $\|\hat{y} - y\|_2$ (in log scale).

Moreover, since the model parameters are shared across both learning and memorization gradient computations, the standard deviations of all weights remain unchanged for both cases. Consequently, the term, $\sigma_{out}\left(\sqrt{d_{out}} + \sqrt{d_1}\right) \cdot \left[\prod_{j=i}^N \left\{ \left(1 + \frac{\sigma_{W_{1,j}} \sigma_{W_{2,j}}}{\sigma_{\tilde{x}_j}} C_{\text{ffn}}\right) \cdot \left(1 + \frac{\sigma_{W_{O,j}}}{\sigma_x} C_{\text{attn}}^j\right)\right\}\right]$, in equation 3 in Theorem 1, remains approximately the same for both memorization and learning gradient norm. Hence, the difference between the upper bounds for $\|g_{x_i}^{\text{learn}}\|_2$ and $\|g_{x_i}^{\text{mem}}\|_2$ (likewise for $\|g_{\tilde{x}_i}^{\text{learn}}\|_2$ and $\|g_{\tilde{x}_i}^{\text{mem}}\|_2$) primarily arises due to the prediction error term, $\|\hat{y} - y\|_2$. In particular, as the model is highly overfitted, during inference on a memorized sample $(x^{\text{NL}}, y^{\text{NL}})$, it confidently predicts the noisy label. As a result, the softmax output $\hat{y}^{\text{mem}}$ places nearly all of its probability mass on the noisy class $y^{\text{NL}}$, i.e., $\hat{y}^{\text{mem}} \approx y^{\text{NL}}$. Therefore, $\|\hat{y}^{\text{mem}} - y^{\text{NL}}\|_2$ is close to zero. In contrast, for a clean test sample $(x^c, y^c)$, the model has acquired generalizable features for class $c$, but has also been exposed to noisy label instances during training, which (when overfitted) prevents ideal generalization (i.e., perfect test accuracy). As a result, the prediction $\hat{y}^{\text{learn}}$ is not sharply peaked at class $y^c$, and instead distributes some probability mass across incorrect classes. Consequently, the prediction error $\|\hat{y}^{\text{learn}} - y^c\|_2$ is noticeably larger, leading to a higher upper bound—and typically, a larger actual learning gradient norm—compared to the memorization case. This behavior, where the prediction error for memorization samples is smaller than that for learning samples, is also verified empirically, as shown in Fig. 7 for TinyViT. Similar trends are observed across other architectures, including GPT2-Small, GPT2-Medium, Smol-LM, Qwen2, ViT-Base, BEiT, and DeiT, as mentioned in Appendix E.5. As a result, this consistent gap in prediction error results in the following relation:

$$\|g_{x_i}^{\text{learn}}\|_2 > \|g_{x_i}^{\text{mem}}\|_2 \;\&\; \|g_{\tilde{x}_i}^{\text{learn}}\|_2 > \|g_{\tilde{x}_i}^{\text{mem}}\|_2 \quad \text{across all layers.} \quad (5)$$

Please note that an ideal case of perfect learning, where 100% memorization and 100% learning co-exist, is not achievable in practice as memorization inherently hinders generalization. Hence, the equality ($\|g_{x_i}^{\text{learn}}\|_2 = \|g_{x_i}^{\text{mem}}\|_2$ and $\|g_{\tilde{x}_i}^{\text{learn}}\|_2 = \|g_{\tilde{x}_i}^{\text{mem}}\|_2$) can be disregarded in almost all cases.

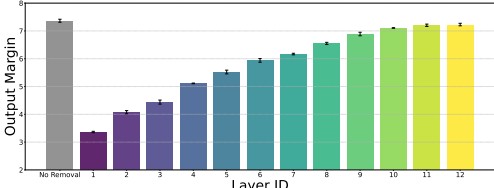 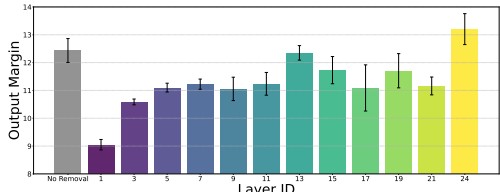

(a) Output Margins after removing residual connections across layers. (CIFAR100-TinyViT)

(b) Output Margins after removing residual connections across layers. (Emotions-GPT2-Medium)

Figure 8: **Output margin corroborates the importance of early residuals.** Removing residual connections from early layers drastically reduces the output margin, increasing uncertainty and misclassifications. In contrast, removing later residuals has a smaller effect—highlighting that early residuals play a crucial role in enabling confident learning.

To summarize, both theoretical and empirical evidence coincide to explain why learning gradients dominate memorization gradients. Theoretically, Theorem 1 explains that the gradient norm is governed primarily by the prediction error term $\|\hat{y} - y\|_2$, which is inherently smaller for memorized samples as seen in Fig. 7. Empirically, this difference translates to consistently smaller memorization gradient norms in comparison to learning gradient norms, as seen in Figs. 5a, 5b.

## 5    EARLY RESIDUALS ARE CRITICAL FOR LEARNING

### 5.1    EARLY RESIDUALS IMPACT ON ACCURACY

In Sec. 4, we revealed how and why residual connections do not contribute to memorization but play a critical role in enabling learning. Now we further investigate the details of this phenomenon, where we analyze the layer-wise impact of residual connections on learning performance. From Figs. 2b & 2d, it is clearly observed that removing residual connections from early layers substantially impairs learning. Consistent results are observed for higher noise level ratios as shown in Figs. 3a & 3b and for the generative modeling tasks presented in Figs. 4a & 4b. Formally, removing the residual connections from the $i^{\text{th}}$ layer leads to lower test accuracy than removing it from the $(i + 1)^{\text{th}}$ layer, i.e., $\text{Acc}(\text{Res}_i) \leq \text{Acc}(\text{Res}_{i+1}), \forall\, 1 \leq i < N$. Consistent trends are also observed across various architectures, including GPT2-Medium, Smol-LM, Qwen2, DeiT, BEiT, and TinyViT, as detailed in Appendix E.1. This finding is also supported in part by prior works (Gromov et al., 2024; Li et al., 2024; Lad et al., 2024; Men et al., 2024), which demonstrated the importance of early layers from a coarse-grained view but not specifically for residual connections nor memorization. However, our study provides a distinctive observation by isolating the impact of residual flow on memorization and learning in each layer.

### 5.2    EXPLAINING THE IMPORTANCE OF EARLY RESIDUALS FOR LEARNING

In Sec. 5, we observed that residual connections in early layers are especially critical for learning in Transformers. A natural question that arises is: *why are early residuals more important than those in later layers?* To answer this, we examine three metrics: **gradient norms**, **standard deviation of residual connections**, and **output margins**.

#### 5.2.1    GRADIENT NORM ANALYSIS

First, we analyze the learning gradient norm, $\|g^{\text{learn}}\|_2$, layer by layer. As shown in Figs. 5a & 5b, we find that early layers exhibit significantly higher gradient magnitudes than later layers. This indicates that the residual connections in early layers are more actively involved in propagating meaningful learning signals, in comparison to later residuals. This provides us with a plausible explanation of why removing early residuals would lead to a significant drop in the test accuracy as seen in Figs. 2b & 2d. Similar trends are observed for other models, GPT2-Small, Smol-LM, Qwen2, ViT-Base, BEiT, and DeiT, as shown in Appendix E.6. We also did the gradient norm analysis across epochs and observed a consistent trend that early residuals exhibit higher norms than middle/later ones during the course of training, as shown in Appendix E.10.1.

To explain the observed empirical phenomenon, we provide a layer-wise theoretical analysis of the upper bound of the gradient norm, as defined in Theorem 1. In the expression, the gradient norm at any layer depends on several components, including the prediction error, classification

head, FFN weights, MHSA weights, and residual connections standard deviations. When comparing two consecutive layers, the prediction error and classification head terms remain constant. Meanwhile, each layer introduces additional multiplicative terms of the form, $(1 + A)(1 + B)$, where $A = \frac{\sigma_{W_{1,j}} \sigma_{W_{2,j}}}{\sigma_{\tilde{x}_j}} C_{\text{ffn}}$, & $B = \frac{\sigma_{W_{O,j}}}{\sigma_{x_j}} C_{\text{attn}}^j$. Here, clearly $A \geq 0$ & $B \geq 0$, since they are composed of standard deviations, where $\sigma_{W_{1,j}}, \sigma_{W_{2,j}}, \sigma_{W_{O,j}}, \sigma_{\tilde{x}_j}, \sigma_{x_j} \geq 0$, and $C_{\text{ffn}} > 0, C_{\text{attn}}^j \geq 0$ (proof provided in Appendix C.). Therefore, as these $(1+A)(1+B)$ multiplicative factors accumulate through the layers, early layers experience a compounded effect, which pushes their gradient norms upper bound higher in comparison to later layers. Hence, building upon this formulation, we establish in Theorem 2 that the gradient norm's upper bound decays with depth, thereby providing a theoretical explanation for why early residuals tend to exhibit larger gradient norm than later residuals, as observed in Figs. 5a & 5b.

> **Theorem 2** (**Upper bound of the gradient norm of early layers residuals are higher than that of later layers residuals**). *It is formally represented as follows:*
>
> $$\text{UB}(\|g_{x_1}\|_2) \geq \text{UB}(\|g_{x_2}\|_2) \geq \cdots \geq \text{UB}(\|g_{x_N}\|_2) \tag{6}$$
>
> *where $UB(\|g_{x_i}\|_2)$ denotes the upper bound of $\|g_{x_i}\|_2$, and $x_i$ is the input to the $i^{th}$ layer's first residual block.*

A formal proof of Theorem 2 is provided in Appendix B.

### 5.2.2 STANDARD DEVIATION ANALYSIS

While Theorem 2 explains the depth-wise decay in gradient norm upper bounds through multiplicative $(1 + A)(1 + B)$ terms, the influence of residual connections and model parameters statistics on this decay remains unresolved. Since both A and B are defined in terms of various standard deviations, understanding their behavior across layers is crucial. Therefore, we now empirically examine the layer-wise variation of statistics of (i) FFN weights ($\sigma_{W_{1,j}}, \sigma_{W_{2,j}}$), (ii) MHSA weights ($\sigma_{W_{O,j}}, C_{\text{attn}}^j$) statistics, and (iii) residual connection inputs ($\sigma_{x_j}, \sigma_{\tilde{x}_j}$). This helps reveal which statistics dominate the $(1 + A)(1 + B)$ terms and thus influence the observed gradient norm pattern across depths through Figs. 5a & 5b.

To understand how different standard deviations evolve across the network, we begin by analyzing the standard deviation of these values across the network. As shown in Table 6a, the standard deviations of model parameters, $\sigma_{W_{j,1}}, \sigma_{W_{j,2}}$, & $\sigma_{W_{j,O}}$, remain relatively stable across layers (similarly, we also show that $C_{\text{attn}}^j$ remains stable across layers in Appendix E.7.1.), as indicated by their low variability. In contrast, the statistics of the residual connections, $\sigma_{x_j}$ and $\sigma_{\tilde{x}_j}$, exhibit significant variation, suggesting greater sensitivity to layer depth. This motivates a deeper, layer-wise investigation into the behavior of residual connection statistics. Figs. 6b & 6c show that the standard deviations of the residual connection inputs, $\sigma_{x_j}$ and $\sigma_{\tilde{x}_j}$, are substantially smaller in early layers compared to later ones. Now, according to equation 3 in Theorem 1, the upper bound of the gradient norm has an inverse relation with the standard deviations of the residual connections. Consequently, smaller residual standard deviations in early layers lead to larger upper bounds of gradient norm. Thus, our empirical analysis reveals that **standard deviation of residual connections**, is a pivotal factor that further promotes higher gradient norms in the early layers compared to the later ones as observed in Figs. 5a & 5b. Similar observations are also seen across other models, GPT2-Small, Smol-LM, Qwen2, ViT-Base, BEiT, and DeiT, as presented in Appendix E.7.

### 5.2.3 OUTPUT MARGIN ANALYSIS

Next, we analyze the model's **output margins**—defined as the difference between the largest and second-largest predicted logits for a sample (Jiang et al., 2018). This margin reflects the confidence in the model's predictions and can serve as a useful proxy for the distance to the decision boundary. In our study, we compare the average output margin across all test samples before and after removing residual connections. As shown in Figs. 8a & 8b, the output margins are substantially reduced when early residuals are removed, whereas they remain relatively stable when later residuals are ablated. Smaller margins indicate that predictions are closer to the decision boundary and thus more prone to misclassification, which explains the observed drop in accuracy following early residuals removal. Consistent observations are seen across other models, GPT2-Small, Smol-LM, Qwen2, ViT-Base, BEiT, and DeiT, in Appendix E.8. We also provide the output margin analysis across epochs in Appendix E.10.2.

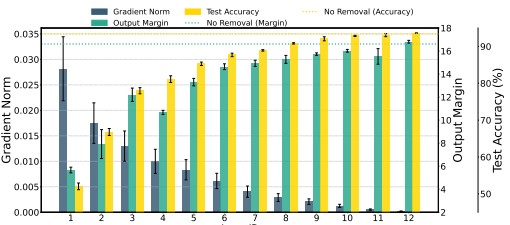 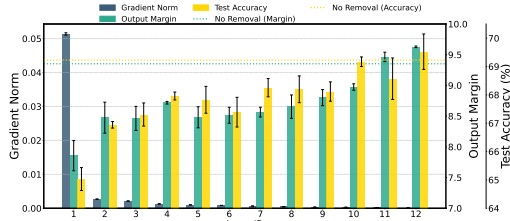

(a) Relation between gradient norm $\|g^{\text{learn}}\|_2$, output margin, and testing accuracy. (CIFAR10-ViT-Base)

(b) Relation between gradient norm $\|g^{\text{learn}}\|_2$, output margin, and testing accuracy. (20NewsGroup-GPT2-Small)

Figure 9: **Gradient norm correlates with output margin and test accuracy.** Residual connections with higher $\|g^{\text{learn}}\|_2$ (early residuals) yield lower output margins and degraded accuracy when removed, indicating their greater importance for learning. Conversely, residuals with lower $\|g^{\text{learn}}\|_2$ (later residuals) have minimal effect on margin and accuracy when removed.

### 5.2.4 UNIFIED VIEW OF GRADIENT NORM, OUTPUT MARGIN AND TEST ACCURACY

Together, these findings, grounded by both empirical analysis and theoretical support, paint a clear picture of why early residuals play a significant role. They carry the strongest learning signals, as seen from both their measured and bounded gradient norms. In comparison to later layers, their role is substantially more critical: when early residuals are removed, the model's output margins and prediction confidence drop, leading to higher prediction errors and ultimately reduced test accuracy.

To further validate this connection, we visualize the relationship between these three measures: (i) gradient norm, $\|g^{\text{learn}}\|_2$, (ii) output margin, and (iii) test accuracy. As shown in Figs. 9a & 9b, residual connections with higher gradient norms cause more drops in output margin, and thus are more subject to test accuracy drops when removed. In contrast, since residual connections with lower gradient norms have less potential, hence, even if they are removed, the impact is insignificant on the output margin and test accuracy. The trend in Fig. 9 can be formally expressed as follows:

$$
\begin{aligned}
\text{High } \|g^{\text{learn}}_{x_\ell}\|_2 &\implies \text{Low Output Margin}^{\text{removed}}_{\text{Res}_\ell} \implies \text{Low Acc}^{\text{removed}}_{\text{Res}_\ell} \\
\text{Low } \|g^{\text{learn}}_{x_\ell}\|_2 &\implies \text{High Output Margin}^{\text{removed}}_{\text{Res}_\ell} \implies \text{High Acc}^{\text{removed}}_{\text{Res}_\ell}
\end{aligned}
\tag{7}
$$

Here, High $\|g^{\text{learn}}_{x_\ell}\|_2$ indicates that the residual connection at layer $\ell$—typically from earlier layers—has a high learning gradient norm. Removing such a residual causes a substantial drop in output margin (Low Output Margin$^{\text{removed}}_{\text{Res}_\ell}$), leading to increased prediction error and reduced test accuracy (Low Acc$^{\text{removed}}_{\text{Res}_\ell}$). Conversely, Low $\|g^{\text{learn}}_{x_\ell}\|_2$—often observed in later residuals—suggests that later residual connections contribute less to learning. Removing these residuals tends to minimally impact the output margin (High Output Margin$^{\text{removed}}_{\text{Res}_\ell}$) and the test accuracy (High Acc$^{\text{removed}}_{\text{Res}_\ell}$), relative to early residuals removal. This contrast highlights a clear relation: **early residuals with high gradient norms are essential for preserving model confidence and generalization, while later residuals with low gradient norms play a more limited role**. Consistent observations are observed for all the other models, GPT2-Medium, Smol-LM, Qwen2, TinyViT, BEiT, and DeiT, in Appendix E.9.

## 6 CONCLUSION

In this work, we show that residual connections in transformers only relay generalization but not memorization, where their removal only impairs learning. We further explain this phenomenon via gradients analysis where memorization gradient norms are much smaller than learning gradient norms across all layers, indicating limited flow of memorization related signal through residual paths. On top of that, we also emphasize the importance of early residuals towards learning where early residuals have higher gradient norms, and their removal causes a larger drop in output margins and test accuracy compared to later residuals. Overall, our findings uncover a novel, key insight in which residual connections only transfer generalization while skipping memorization.

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

APPENDIX

## A PROOF OF THEOREM 1

**Theorem 1: Upper Bound of Gradient Norm** *Let $x_i$ be the input to the $i^{th}$ layer's first residual block. Then, the gradient norm satisfies:*

$$\|g_{x_i}\|_2 = \left\|\frac{\partial \mathcal{L}}{\partial x_i}\right\|_2 \le \underbrace{\|\hat{y} - y\|_2}_{\text{error}} \cdot \sigma_{\text{out}} \left(\sqrt{d_{\text{out}}} + \sqrt{d_1}\right)$$

$$\cdot \left[\prod_{j=i}^{N}\left\{\left(1 + \frac{\sigma_{W_{1,j}} \sigma_{W_{2,j}}}{\sigma_{\tilde{x}_j}} C_{\text{ffn}}\right) \cdot \left(1 + \frac{\sigma_{W_{O,j}}}{\sigma_{x_j}} C_{\text{attn}}^{j}\right)\right\}\right] \tag{8}$$

*where $y$ is the ground truth one-hot encoded vector, $\hat{y}$ is the predicted softmax probability vector, $\sigma_{x_j}$ and $\sigma_{\tilde{x}_j}$ are the standard deviations of the residual stream inputs $x_j$ and $\tilde{x}_j$ respectively, $\sigma_{W_{1,j}}, \sigma_{W_{2,j}}$ are the standard deviations of the feedforward network (FFN) weight matrices $W_{1,j}$, $W_{2,j}$ respectively, $\sigma_{W_{O,j}}$ is the standard deviation of the output projection matrix in the MHSA block for $j^{th}$ layer, $C_{ffn} = \left(\sqrt{d_1} + \sqrt{d_2}\right)^2$ where $d_1, d_2$ are intermediate hidden sizes, $C_{attn}^{j} = 2d_1 \cdot \left\|J_Z^j\right\|_2$, $\sigma_{out}$ is the standard deviation of classification head weight matrix $W_{out}$, and $d_{out}$ is output size of classification head.*

***Proof:***

Firstly, we formally describe the multi-head self-attention and feed-forward blocks in the transformer architecture as follows:

$$\tilde{x}_i = x_i + \text{MHSA}(\text{LN}(x_i)) \quad \& \quad y_i = \tilde{x}_i + \text{FFN}(\text{LN}(\tilde{x}_i)) \tag{9}$$

where $x_i$ denotes the input to the first residual block; $\tilde{x}_i$ denotes the output of the first residual block and is also the input of the second residual block; $y_i$ is the output of the second residual block in the $i^{\text{th}}$ transformer layer. MHSA, FFN, and LN denote the multi-head self-attention, feedforward, and LayerNorm layers, respectively. Since there are 2 residual blocks, we compute the gradient of loss with respect to the inputs of both of them separately, which is, $x_i$ and $\tilde{x}_i$.

### A.1 GRADIENT ANALYSIS FOR SECOND RESIDUAL CONNECTION $\tilde{x}_i$ ($i.e., g_{\tilde{x}_i}$)

The gradient $g_{\tilde{x}_i}$ for the second residual connection $\tilde{x}_i$, can be written as follows:

$$g_{\tilde{x}_i} = \frac{\partial \mathcal{L}}{\partial \tilde{x}_i} = \frac{\partial \mathcal{L}}{\partial y_{\text{out}}} \cdot \frac{\partial y_{\text{out}}}{\partial y_N} \cdot \prod_{j=i+1}^{N}\left(\frac{\partial y_j}{\partial \tilde{x}_j} \cdot \frac{\partial \tilde{x}_j}{\partial x_j}\right) \cdot \frac{\partial y_i}{\partial \tilde{x}_i} \tag{10}$$

where $\mathcal{L}$ is the cross entropy loss, $y_N$ is the output of the $N^{\text{th}}$ transformer layer, $y_{\text{out}}$ is the output of the classification layer, and $y_i = x_{i+1}$ because $i^{\text{th}}$ layer's output $y_i$ is the input of $(i+1)^{\text{th}}$ layer, $x_{i+1}$. The cross entropy loss $\mathcal{L}$ between the ground truth vector $y$ and the predicted softmax probability vector $\hat{y}$ ($= \text{Softmax}(y_{\text{out}})$), is written as follows:

$$\mathcal{L} = -\sum_{c=1}^{C} y_c \log(\hat{y}_c) \tag{11}$$

where $y_c = 1$ if $c$ is the ground truth class, otherwise 0, and $\hat{y}_c$ is the predicted softmax probability of class $c$. We can then write $\frac{\partial \mathcal{L}}{\partial y_{\text{out}}}$ as follows:

$$\frac{\partial \mathcal{L}}{\partial y_{\text{out}}} = \hat{y} - y \tag{12}$$

We then substitute equation 12 to equation 10 and obtain the following:

$$g_{\tilde{x}_i} = \frac{\partial \mathcal{L}}{\partial \tilde{x}_i} = (\hat{y} - y) \cdot \frac{\partial y_{\text{out}}}{\partial y_N} \cdot \prod_{j=i+1}^{N} \left( \frac{\partial y_j}{\partial \tilde{x}_j} \cdot \frac{\partial \tilde{x}_j}{\partial x_j} \right) \cdot \frac{\partial y_i}{\partial \tilde{x}_i} \tag{13}$$

We now take the $\ell_2$ norm on both sides of equation 13 to get the following.

$$\|g_{\tilde{x}_i}\|_2 = \left\| \frac{\partial \mathcal{L}}{\partial \tilde{x}_i} \right\|_2 = \left\| (\hat{y} - y) \cdot \frac{\partial y_{\text{out}}}{\partial y_N} \cdot \prod_{j=i+1}^{N} \left( \frac{\partial y_j}{\partial \tilde{x}_j} \cdot \frac{\partial \tilde{x}_j}{\partial x_j} \right) \cdot \frac{\partial y_i}{\partial \tilde{x}_i} \right\|_2 \tag{14}$$

From the Cauchy–Schwarz inequality (Steele, 2004) (via the multiplicative property of the operator norm), we know that:

$$\|A_1 A_2 \cdots A_n\|_2 \leq \|A_1\|_2 \cdot \|A_2\|_2 \cdots \|A_n\|_2 \tag{15}$$

where $A_i$ are matrices/vectors. Accordingly, after applying equation 15 to equation 14, we obtain the following:

$$\|g_{\tilde{x}_i}\|_2 = \left\| \frac{\partial \mathcal{L}}{\partial \tilde{x}_i} \right\|_2 = \left\| (\hat{y} - y) \cdot \frac{\partial y_{\text{out}}}{\partial y_N} \cdot \prod_{j=i+1}^{N} \left( \frac{\partial y_j}{\partial \tilde{x}_j} \cdot \frac{\partial \tilde{x}_j}{\partial x_j} \right) \cdot \frac{\partial y_i}{\partial \tilde{x}_i} \right\|_2$$
$$\leq \|\hat{y} - y\|_2 \cdot \left\| \frac{\partial y_{\text{out}}}{\partial y_N} \right\|_2 \cdot \left[ \prod_{j=i+1}^{N} \left( \left\| \frac{\partial y_j}{\partial \tilde{x}_j} \right\|_2 \cdot \left\| \frac{\partial \tilde{x}_j}{\partial x_j} \right\|_2 \right) \right] \cdot \left\| \frac{\partial y_i}{\partial \tilde{x}_i} \right\|_2 \tag{16}$$

We know that, $y_{\text{out}} = W_{\text{out}} * y_N$, where $W_{\text{out}}$ is the weight matrix of the classification head. Similarly, from equation 2, we know that $\tilde{x}_j = x_j + \text{MHSA}(\text{LN}(x_j))$ and $y_j = \tilde{x}_j + \text{FFN}(\text{LN}(\tilde{x}_j))$. Also, from Takase et al. (2023), we already know the upper bounds of the $\ell_2$-norms of $\frac{\partial y_{\text{out}}}{\partial y_N}$, $\frac{\partial y_j}{\partial \tilde{x}_j}$, and $\frac{\partial \tilde{x}_j}{\partial x_j}$, as follows:

$$\text{UB}\left( \left\| \frac{\partial y_{\text{out}}}{\partial y_N} \right\|_2 \right) = \sigma_{\text{out}} \left( \sqrt{d_{\text{out}}} + \sqrt{d_1} \right), \quad \text{UB}\left( \left\| \frac{\partial y_j}{\partial \tilde{x}_j} \right\|_2 \right) = 1 + \frac{\sigma_{W_{O,j}}}{\sigma_{x_j}} C_{\text{attn}}^j,$$
$$\text{UB}\left( \left\| \frac{\partial \tilde{x}_j}{\partial x_j} \right\|_2 \right) = 1 + \frac{\sigma_{W_{1,j}} \sigma_{W_{2,j}}}{\sigma_{\tilde{x}_j}} C_{\text{ffn}}, \tag{17}$$

where, $\sigma_{x_j}$ and $\sigma_{\tilde{x}_j}$ are the standard deviations of the residual stream inputs $x_j$ and $\tilde{x}_j$, respectively, $\sigma_{W_{1,j}}, \sigma_{W_{2,j}}$ are the standard deviations of the feedforward network (FFN) weight matrices $W_{1,j}$, $W_{2,j}$, respectively, $\sigma_{W_{O,j}}$ is the standard deviation of the output projection matrix in the MHSA block for $j^{\text{th}}$ layer, $C_{\text{ffn}} = \left( \sqrt{d_1} + \sqrt{d_2} \right)^2$, $C_{\text{attn}}^j = 2d_1 \cdot \left\| J_Z^j \right\|_2$, $d_1, d_2$ are intermediate hidden sizes, $\sigma_{\text{out}}$ is the standard deviation of classification head weight matrix $W_{\text{out}}$, $d_{\text{out}}$ is output size of classification head, $\left\| J_Z^j \right\|_2 = h \left( \left( \sqrt{L} + 2 + \frac{1}{\sqrt{L}} \right) \sigma_{Q,j}^3 \sqrt{d_1^3 d_{\text{head}}} + \sigma_{Q,j} \left( \sqrt{d_1} + \sqrt{d_{\text{head}}} \right) \right)$ where $\sigma_{Q,j}$ is the standard deviation of attention query matrix, $h$ is the number of attention heads, $d_{\text{head}}$ is size of each attention head, and $L$ is the input sequence length.

Therefore, we replace the gradient norms terms in equation 16 with the terms defined in equation 17, to obtain the following expression:

$$\|g_{\tilde{x}_i}\|_2 = \left\| \frac{\partial \mathcal{L}}{\partial \tilde{x}_i} \right\|_2 \leq \underbrace{\|\hat{y} - y\|_2}_{\text{error}} \cdot \sigma_{\text{out}} \left( \sqrt{d_{\text{out}}} + \sqrt{d_1} \right)$$
$$\cdot \left[ \prod_{j=i+1}^{N} \left\{ \left( 1 + \frac{\sigma_{W_{O,j}}}{\sigma_{x_j}} C_{\text{attn}}^j \right) \left( 1 + \frac{\sigma_{W_{1,j}} \sigma_{W_{2,j}}}{\sigma_{\tilde{x}_j}} C_{\text{ffn}} \right) \right\} \right] \left( 1 + \frac{\sigma_{W_{O,i}}}{\sigma_{x_i}} C_{\text{attn}}^i \right) \tag{18}$$

$$\|g_{\tilde{x}_i}\|_2 = \left\| \frac{\partial \mathcal{L}}{\partial \tilde{x}_i} \right\|_2 \leq \underbrace{\|\hat{y} - y\|_2}_{\text{error}} \cdot \sigma_{\text{out}} \left( \sqrt{d_{\text{out}}} + \sqrt{d_1} \right)$$
$$\cdot \left[ \prod_{j=i}^{N} \left( 1 + \frac{\sigma_{W_{O,j}}}{\sigma_{x_j}} C_{\text{attn}}^j \right) \right] \cdot \left[ \prod_{j=i+1}^{N} \left( 1 + \frac{\sigma_{W_{1,j}} \sigma_{W_{2,j}}}{\sigma_{\tilde{x}_j}} C_{\text{ffn}} \right) \right] \tag{19}$$

## A.2 Gradient Analysis for first residual connection $x_i$ (i.e., $g_{x_i}$):

Similar to $g_{\tilde{x}_i}$, we can represent the gradient norm for the first residual connection $x_i$, i.e., $g_{x_i}$ as follows:

$$g_{x_i} = \frac{\partial \mathcal{L}}{\partial x_i} = \frac{\partial \mathcal{L}}{\partial y_{\text{out}}} \cdot \frac{\partial y_{\text{out}}}{\partial y_N} \cdot \prod_{j=i+1}^{N} \left( \frac{\partial y_j}{\partial \tilde{x}_j} \cdot \frac{\partial \tilde{x}_j}{\partial x_j} \right) \cdot \frac{\partial y_i}{\partial \tilde{x}_i} \cdot \frac{\partial \tilde{x}_i}{\partial x_i} \tag{20}$$

In that, we substitute $\frac{\partial \mathcal{L}}{\partial y_{\text{out}}}$ with equation 12, as follows:

$$g_{x_i} = \frac{\partial \mathcal{L}}{\partial x_i} = (\hat{y} - y) \cdot \frac{\partial y_{\text{out}}}{\partial y_N} \cdot \prod_{j=i+1}^{N} \left( \frac{\partial y_j}{\partial \tilde{x}_j} \cdot \frac{\partial \tilde{x}_j}{\partial x_j} \right) \cdot \frac{\partial y_i}{\partial \tilde{x}_i} \cdot \frac{\partial \tilde{x}_i}{\partial x_i} \tag{21}$$

We then apply the $\ell_2$ norm on both sides of the Eq. equation 21,

$$\|g_{x_i}\|_2 = \left\| \frac{\partial \mathcal{L}}{\partial x_i} \right\|_2 = \left\| (\hat{y} - y) \cdot \frac{\partial y_{\text{out}}}{\partial y_N} \cdot \prod_{j=i+1}^{N} \left( \frac{\partial y_j}{\partial \tilde{x}_j} \cdot \frac{\partial \tilde{x}_j}{\partial x_j} \right) \cdot \frac{\partial y_i}{\partial \tilde{x}_i} \cdot \frac{\partial \tilde{x}_i}{\partial x_i} \right\|_2 \tag{22}$$

We then apply the Cauchy–Schwarz inequality (Steele, 2004), as shown in equation 15, to obtain the following:

$$\|g_{x_i}\|_2 = \left\| \frac{\partial \mathcal{L}}{\partial x_i} \right\|_2 \leq \|\hat{y} - y\|_2 \left\| \frac{\partial y_{\text{out}}}{\partial y_N} \right\|_2 \cdot \left[ \prod_{j=i+1}^{N} \left( \left\| \frac{\partial y_j}{\partial \tilde{x}_j} \right\|_2 \cdot \left\| \frac{\partial \tilde{x}_j}{\partial x_j} \right\|_2 \right) \right] \cdot \left\| \frac{\partial y_i}{\partial \tilde{x}_i} \right\|_2 \cdot \left\| \frac{\partial \tilde{x}_i}{\partial x_i} \right\|_2 \tag{23}$$

We now, expand the $\ell_2$-norms of the gradients using equation 17 to obtain the following expression:

$$\|g_{x_i}\|_2 = \left\| \frac{\partial \mathcal{L}}{\partial x_i} \right\|_2 \leq \underbrace{\|\hat{y} - y\|_2}_{\text{error}} \cdot \sigma_{\text{out}} \left( \sqrt{d_{\text{out}}} + \sqrt{d_1} \right)$$
$$\cdot \left[ \prod_{j=i+1}^{N} \left\{ \left( 1 + \frac{\sigma_{W_{O,j}}}{\sigma_{x_j}} C_{\text{attn}}^j \right) \left( 1 + \frac{\sigma_{W_{1,j}} \sigma_{W_{2,j}}}{\sigma_{\tilde{x}_j}} C_{\text{ffn}} \right) \right\} \right] \tag{24}$$
$$\cdot \left( 1 + \frac{\sigma_{W_{O,i}}}{\sigma_{x_i}} C_{\text{attn}}^i \right) \cdot \left( 1 + \frac{\sigma_{W_{1,i}} \sigma_{W_{2,i}}}{\sigma_{\tilde{x}_i}} C_{\text{ffn}} \right)$$

$$\|g_{x_i}\|_2 = \left\| \frac{\partial \mathcal{L}}{\partial x_i} \right\|_2 \leq \underbrace{\|\hat{y} - y\|_2}_{\text{error}} \cdot \sigma_{\text{out}} \left( \sqrt{d_{\text{out}}} + \sqrt{d_1} \right)$$
$$\cdot \left[ \prod_{j=i}^{N} \left\{ \left( 1 + \frac{\sigma_{W_{O,j}}}{\sigma_{x_j}} C_{\text{attn}}^j \right) \cdot \left( 1 + \frac{\sigma_{W_{1,j}} \sigma_{W_{2,j}}}{\sigma_{\tilde{x}_j}} C_{\text{ffn}} \right) \right\} \right] \tag{25}$$

In conclusion, the $\ell_2$ norm of the gradient of the loss $\mathcal{L}$ w.r.t the input of each of the residual blocks, $\tilde{x}_i$ and $x_i$, is upper bounded as shown in equation 19 and equation 25, respectively.

$\square$

## B Proof of Theorem 2

**Theorem 2: Upper bound of the gradient norm of early layers residuals are higher than that of later layers residuals.** *It is formally represented as follows:*

$$\text{UB}(\|g_{x_1}\|_2) \geq \text{UB}(\|g_{x_2}\|_2) \geq \cdots \geq \text{UB}(\|g_{x_N}\|_2) \tag{26}$$

*where $\text{UB}(\|g_{x_i}\|_2)$ denotes the upper bound of $\|g_{x_i}\|_2$, and $x_i$ is the input to the $i^{th}$ layer's first residual block.*

*Proof:*

We utilize the derived upper bound of the $\ell_2$-norm of the loss gradient w.r.t. each of the 2 residual block inputs, $\tilde{x}_i$ and $x_i$ in equation 19 and equation 25, respectively.

## B.1 UPPER BOUND ANALYSIS FOR SECOND RESIDUAL CONNECTION'S GRADIENT NORM $\|g_{\tilde{x}_i}\|_2$

To analyze how the gradient norms behave across layers, we compare the upper bounds of the gradient norms of the second residual connection, for 2 consecutive layers, $i$ and $i+1$, $\|g_{\tilde{x}_i}\|_2$ and $\|g_{\tilde{x}_{i+1}}\|_2$. Accordingly, from Theorem 1, the upper bound of $\|g_{\tilde{x}_i}\|_2$ and $\|g_{\tilde{x}_{i+1}}\|_2$ can be written as:

$$
\begin{aligned}
\text{UB}\left(\|g_{\tilde{x}_i}\|_2\right) = \|\hat{y} - y\|_2 \cdot \sigma_{\text{out}} \left(\sqrt{d_{\text{out}}} + \sqrt{d_1}\right) \\
\cdot \left[\prod_{j=i}^{N}\left(1 + \frac{\sigma_{W_{O,j}}}{\sigma_{x_j}} C_{\text{attn}}^j\right)\right] \cdot \left[\prod_{j=i+1}^{N}\left(1 + \frac{\sigma_{W_{1,j}} \sigma_{W_{2,j}}}{\sigma_{\tilde{x}_j}} C_{\text{ffn}}\right)\right]
\end{aligned}
\tag{27}
$$

$$
\begin{aligned}
\text{UB}\left(\|g_{\tilde{x}_{i+1}}\|_2\right) = \|\hat{y} - y\|_2 \cdot \sigma_{\text{out}} \left(\sqrt{d_{\text{out}}} + \sqrt{d_1}\right) \\
\cdot \left[\prod_{j=i+1}^{N}\left(1 + \frac{\sigma_{W_{O,j}}}{\sigma_{x_j}} C_{\text{attn}}^j\right)\right] \cdot \left[\prod_{j=i+2}^{N}\left(1 + \frac{\sigma_{W_{1,j}} \sigma_{W_{2,j}}}{\sigma_{\tilde{x}_j}} C_{\text{ffn}}\right)\right]
\end{aligned}
\tag{28}
$$

We now check if $\text{UB}(\|g_{\tilde{x}_i}\|_2) \geq \text{UB}(\|g_{\tilde{x}_{i+1}}\|_2)$ from equation 27 and equation 28 respectively, as follows:

$$
\begin{aligned}
\|\hat{y} - y\|_2 \cdot \sigma_{\text{out}} \left(\sqrt{d_{\text{out}}} + \sqrt{d_1}\right) \cdot \left[\prod_{j=i}^{N}\left(1 + \frac{\sigma_{W_{O,j}}}{\sigma_{x_j}} C_{\text{attn}}^j\right)\right] \cdot \left[\prod_{j=i+1}^{N}\left(1 + \frac{\sigma_{W_{1,j}} \sigma_{W_{2,j}}}{\sigma_{\tilde{x}_j}} C_{\text{ffn}}\right)\right] \\
\geq \|\hat{y} - y\|_2 \cdot \sigma_{\text{out}} \left(\sqrt{d_{\text{out}}} + \sqrt{d_1}\right) \cdot \left[\prod_{j=i+1}^{N}\left(1 + \frac{\sigma_{W_{O,j}}}{\sigma_{x_j}} C_{\text{attn}}^j\right)\right] \cdot \left[\prod_{j=i+2}^{N}\left(1 + \frac{\sigma_{W_{1,j}} \sigma_{W_{2,j}}}{\sigma_{\tilde{x}_j}} C_{\text{ffn}}\right)\right]
\end{aligned}
\tag{29}
$$

After further reducing the inequality, we obtain the following:

$$
\left(1 + \frac{\sigma_{W_{O,i}}}{\sigma_{x_i}} C_{\text{attn}}^i\right) \cdot \left(1 + \frac{\sigma_{W_{1,i+1}} \sigma_{W_{2,i+1}}}{\sigma_{\tilde{x}_{i+1}}} C_{\text{ffn}}\right) \geq 1.
\tag{30}
$$

We know that all the standard-deviation terms, $\sigma_{W_{O,i}}$, $\sigma_{x_i}$, $\sigma_{W_{1,i+1}}$, $\sigma_{W_{2,i+1}}$, $\sigma_{\tilde{x}_{i+1}}$, are $\geq 0$ by default. Furthermore, we know that $C_{\text{attn}}^i \geq 0$ and $C_{\text{ffn}} > 0$ as proved in Section C. This means that,

$$
\frac{\sigma_{W_{O,i}}}{\sigma_{x_i}} C_{\text{attn}}^i \geq 0 \quad \& \quad \frac{\sigma_{W_{1,i+1}} \sigma_{W_{2,i+1}}}{\sigma_{\tilde{x}_{i+1}}} C_{\text{ffn}} \geq 0
\tag{31}
$$

This further proves that,

$$
1 + \frac{\sigma_{W_{O,i}}}{\sigma_{x_i}} C_{\text{attn}}^i \geq 1 \quad \& \quad 1 + \frac{\sigma_{W_{1,i+1}} \sigma_{W_{2,i+1}}}{\sigma_{\tilde{x}_{i+1}}} C_{\text{ffn}} \geq 1
\tag{32}
$$

Hence, equation 32, proves that equation 30 holds true, and thereby also proving that $\text{UB}(\|g_{\tilde{x}_i}\|_2) \geq \text{UB}(\|g_{\tilde{x}_{i+1}}\|_2)$ for all $1 \leq i \leq N$.

Therefore, we can conclude that the **upper bound of the gradient norms of the second residual of the early layers is larger than that of the later layers**.

## B.2 UPPER BOUND ANALYSIS FOR FIRST RESIDUAL CONNECTION'S GRADIENT NORM $\|g_{x_i}\|_2$

To analyze how the gradient norms behave across layers, we compare the upper bounds of the gradient norms of the first residual connection, for 2 consecutive layers, $i$ and $i+1$, $\|g_{x_i}\|_2$ and $\|g_{x_{i+1}}\|_2$. Accordingly, from Theorem 1, the upper bound of $\|g_{x_i}\|_2$ and $\|g_{x_{i+1}}\|_2$ can be written as:

$$
\begin{aligned}
\text{UB}\left(\|g_{x_i}\|_2\right) = \|\hat{y} - y\|_2 \cdot \sigma_{\text{out}} \left(\sqrt{d_{\text{out}}} + \sqrt{d_1}\right) \\
\cdot \left[\prod_{j=i}^{N}\left\{\left(1 + \frac{\sigma_{W_{O,j}}}{\sigma_{x_j}} C_{\text{attn}}^j\right) \cdot \left(1 + \frac{\sigma_{W_{1,j}} \sigma_{W_{2,j}}}{\sigma_{\tilde{x}_j}} C_{\text{ffn}}\right)\right\}\right]
\end{aligned}
\tag{33}
$$

$$\text{UB}\left(\|g_{x_{i+1}}\|_2\right) = \|\hat{y} - y\|_2 \cdot \sigma_{\text{out}}\left(\sqrt{d_{\text{out}}} + \sqrt{d_1}\right)$$

$$\cdot \left[\prod_{j=i+1}^{N}\left\{\left(1 + \frac{\sigma_{W_{O,j}}}{\sigma_{x_j}}C_{\text{attn}}^j\right)\cdot\left(1 + \frac{\sigma_{W_{1,j}}\,\sigma_{W_{2,j}}}{\sigma_{\tilde{x}_j}}C_{\text{ffn}}\right)\right\}\right] \quad (34)$$

We now check if $\text{UB}(\|g_{x_i}\|_2) \geq \text{UB}(\|g_{x_{i+1}}\|_2)$ from equation 33 and equation 34 respectively, as follows:

$$\|\hat{y} - y\|_2 \cdot \sigma_{\text{out}}\left(\sqrt{d_{\text{out}}} + \sqrt{d_1}\right)\cdot\prod_{j=i}^{N}\left[\left(1 + \frac{\sigma_{W_{O,j}}}{\sigma_{x_j}}C_{\text{attn}}^j\right)\cdot\left(1 + \frac{\sigma_{W_{1,j}}\,\sigma_{W_{2,j}}}{\sigma_{\tilde{x}_j}}C_{\text{ffn}}\right)\right]$$

$$\geq \|\hat{y} - y\|_2 \cdot \sigma_{\text{out}}\left(\sqrt{d_{\text{out}}} + \sqrt{d_1}\right)\cdot\prod_{j=i+1}^{N}\left[\left(1 + \frac{\sigma_{W_{O,j}}}{\sigma_{x_j}}C_{\text{attn}}^j\right)\cdot\left(1 + \frac{\sigma_{W_{1,j}}\,\sigma_{W_{2,j}}}{\sigma_{\tilde{x}_j}}C_{\text{ffn}}\right)\right]. \quad (35)$$

After further reducing the inequality we obtain the following:

$$\left(1 + \frac{\sigma_{W_{O,i}}}{\sigma_{x_i}}C_{\text{attn}}^i\right)\cdot\left(1 + \frac{\sigma_{W_{1,i}}\,\sigma_{W_{2,i}}}{\sigma_{\tilde{x}_i}}C_{\text{ffn}}\right) \geq 1. \quad (36)$$

We already know that all the standard-deviation terms - $\sigma_{W_{O,i}}$, $\sigma_{x_i}$, $\sigma_{W_{1,i}}$, $\sigma_{W_{2,i}}$, $\sigma_{\tilde{x}_i}$, are $\geq 0$. Furthermore, we know that $C_{\text{attn}}^i \geq 0$ and $C_{\text{ffn}} > 0$ as proved in Section C. This means that,

$$\frac{\sigma_{W_{O,i}}}{\sigma_{x_i}}C_{\text{attn}}^i \geq 0 \quad \& \quad \frac{\sigma_{W_{1,i}}\,\sigma_{W_{2,i}}}{\sigma_{\tilde{x}_i}}C_{\text{ffn}} \geq 0 \quad (37)$$

This further proves that,

$$1 + \frac{\sigma_{W_{O,i}}}{\sigma_{x_i}}C_{\text{attn}}^i \geq 1 \quad \& \quad 1 + \frac{\sigma_{W_{1,i}}\,\sigma_{W_{2,i}}}{\sigma_{\tilde{x}_i}}C_{\text{ffn}} \geq 1 \quad (38)$$

Hence, equation 38, proves that equation 36 holds true, and thereby also proving that $\text{UB}(\|g_{x_i}\|_2) \geq \text{UB}(\|g_{x_{i+1}}\|_2)$ for all $1 \leq i \leq N$.

Therefore, we can conclude that the **upper bound of the gradient norms of the first residual of the early layers is larger than that of the later layers**.

$\square$

## C   Proof for $C_{\text{FFN}} > 0$ and $C_{\text{ATTN}}^j \geq 0$

From Takase et al. (2023), we already know that for each transformer layer $C_{\text{ffn}} = \left(\sqrt{d_1} + \sqrt{d_2}\right)^2$ and $C_{\text{attn}}^j = 2d_1 \cdot \left\|J_Z^j\right\|_2$, where $\left\|J_Z^j\right\|_2 = h\left(\left(\sqrt{L} + 2 + \frac{1}{\sqrt{L}}\right)\sigma_{Q,j}^3\sqrt{d_1^3 d_{\text{head}}} + \sigma_{Q,j}\left(\sqrt{d_1} + \sqrt{d_{\text{head}}}\right)\right)$.

We now need to prove that $C_{\text{ffn}} > 0$ and $C_{\text{attn}}^j \geq 0$. We do that as follows:

We know that for each transformer layer the intermediate hidden sizes, $d_1$ and $d_2$ are $> 0$. Hence, $\left(\sqrt{d_1} + \sqrt{d_2}\right)^2 > 0$. This proves that $C_{\text{ffn}} > 0$.

For $C_{\text{attn}}^j$, we expand it as follows:

$$C_{\text{attn}}^j = 2d_1 h\left(\left(\sqrt{L} + 2 + \frac{1}{\sqrt{L}}\right)\sigma_{Q,j}^3\sqrt{d_1^3 d_{\text{head}}} + \sigma_{Q,j}\left(\sqrt{d_1} + \sqrt{d_{\text{head}}}\right)\right) \quad (39)$$

We know that for any transformer model, the number of attention heads $h > 0$, the size of the attention head $d_{\text{head}} > 0$, the intermediate hidden size $d_1 > 0$, and the standard deviation of the attention query matrix $\sigma_{Q,j} \geq 0$, across all layers. Furthermore, we know that the input length sequence would also be of size at least 1 (assuming that we do not have an empty string as the input). This proves that across all transformer layers,

$$C_{\text{attn}}^j = 2d_1 h\left(\left(\sqrt{L} + 2 + \frac{1}{\sqrt{L}}\right)\sigma_{Q,j}^3\sqrt{d_1^3 d_{\text{head}}} + \sigma_{Q,j}\left(\sqrt{d_1} + \sqrt{d_{\text{head}}}\right)\right) \geq 0 \quad (40)$$

Hence, we have proven that $C_{\text{ffn}} > 0$ and $C_{\text{attn}}^j \geq 0$ across all transformer layers.

# D  TRAINING DETAILS

In this section, we explain the experimental setup of our work, spanning across different vision and language datasets and models used in this study, along with the hyperparameters used to train the models.

## D.1  DATASETS

As part of this study, we considered 7 different datasets covering both vision and language modalities, as follows:

**20NewsGroup** proposed in Lang (1995), is a collection of approximately 20,000 newsgroup documents, partitioned (nearly) evenly across 20 different news groups. We split the dataset into training, validation, and testing using a stratified split of 70:20:10. To induce the notion of noisy labels, we randomly flip labels of 1% proportion of class 1 samples during training, while keeping the rest of the data points the same.

**Emotions** created by Saravia et al. (2018), comprised of 20,000 samples, split across training (16,000), validation (2,000), and testing (2,000). It consists of a total of 6 classes depicting different emotion types. To evaluate memorization, we introduce noisy labels in 1% of the trainset class 5 samples, by changing their labels to a random, different label and keeping the remaining samples unaltered.

**TweetTopic** proposed in Antypas et al. (2022), consists of a collection of social media tweets covering a range of everyday topics. The dataset is split across train (2,858), validation (352), and test (376) sets, consisting of 6 classes. To measure the notion of memorization, we introduce noisy labels, by flipping labels of 1% of class 3 train samples to any other random class label, while keeping rest of the samples the same.

**Places365Mini** is a subset of the standard Places365 dataset originally introduced by Zhou et al. (2017). The Places365Mini version is publicly available on Huggingface[1], and it consists of 8,000 samples spanning across 10 classes, with 7,500 samples in the train set and 500 in the test set. We also resize the images to size 224x224x3 for compatibility with model's input requirements. Furthermore, to study memorization, we induce noisy labels by randomly flipping labels of 1% of class 9 train samples to a different class.

**CIFAR10** proposed by Krizhevsky et al. (2009), comprises of 60,000 samples spanning equally across 10 classes. The training, validation, and testing sets consists of 40,000, 10,000 and 10,000 samples respectively. Furthermore, we resize the images to 224x224x3 for model input requirements. To study memorization, we randomly flip labels of 1% of class 9 train samples to any other random class label.

**CIFAR100** introduced by Krizhevsky et al. (2009), consists of 60,000 samples, spread equally across 100 classes, split across training (40,000), validation (10,000), and testing (10,000) sets. Prior to training, all images are resized to 224×224×3 to match the model's input dimensions. We induce noisy labels, by randomly flipping labels of 1% of class 16 train samples to any other random class label.

**UTK-Face** proposed in Zhang et al. (2017), provides 23,705 face images annotated with 5 ethnicity groups. The dataset is partitioned into training, validation, and testing subsets following a stratified 65:15:20 split to preserve class balance. Before model training, each image is resized to 224×224×3, so that it conforms to the input specifications of the model. To add label noise, we randomly alter the labels of 1% of training samples from class 2, assigning each a label from one of the remaining classes.

Lastly, for each case, to ensure the model memorizes the noisy labels, we train the model till it achieves 100% train accuracy.

---

[1]https://huggingface.co/datasets/dpdl-benchmark/places365-mini-sample-hard

## D.2 MODELS

In this study, we consider 8 transformer models covering both vision and language modalities. We utilize the Sequence Classification variant of these models available on Huggingface[2].

| Language Models | Description |
| --- | --- |
| GPT2-Medium (Radford et al., 2019) | 24-layer unidirectional decoder-only transformer trained for causal language modeling. |
| GPT2-Small (Radford et al., 2019) | 12-layer unidirectional decoder-only transformer trained for causal language modeling. |
| Smol-LM-135m (Allal et al., 2025) | 30-layer small, efficient transformer model developed for easy on-device use. |
| Qwen2-0.5B (Team, 2024) | 24-layer efficient LLM optimized for generative tasks, using RMSNorm. |

| Vision Models | Description |
| --- | --- |
| ViT-Base (Dosovitskiy et al., 2020) | 12-layer Vision Transformer Base model for image classification. |
| TinyViT (Wu et al., 2022) | 12-layer tiny and efficient small vision transformer pretrained on large-scale datasets with a fast distillation framework. |
| BEiT (Bao et al., 2021) | 12-layered transformer that learns rich image representations by predicting masked image patches in a BERT-style self-supervised pretraining framework. |
| DeiT (Touvron et al., 2021) | 12-layer Data-efficient Image Transformer trained with distillation, without external data. |

Table 1: Overview of the 8 language and vision transformer models.

---

[2]https://huggingface.co/docs/transformers/index

### D.3 Training Settings & Hyper-parameters

In our work, we study the following datasets and models setups: (i) Emotions-GPT2-Medium, (ii) 20NewsGroup-GPT2-Small, (iii) TweetTopic-Smol-LM, (iv) TweetTopic-Qwen2, (v) CIFAR10-ViT-Base, (vi) CIFAR100-TinyViT, (vii) Places365Mini-BEiT, and (viii) UTK-Face-DeiT.

In addition to this, we maintain a consistent training setting across all variations. We use Adam as the optimizer and set a learning rate of 2e-5, along with a batch size of 16 for all the models. Then, we train the models for 70 epochs to achieve memorization. In addition to that, we do not use any data augmentation in our training procedures to obscure any impact from augmentations. We used A100, H100 and A5000 GPUs to train our models.

## E  Additional Experiments & Results

In this section, we provide supplementary results for the remaining models for the experiments done in Sec. 4 and Sec. 5. These results further corroborates our contributions - (i) residual connections skip memorization and only relays generalization, (ii) memorization gradient norm is smaller than learning gradient norm across all layers, (iii) memorization gradient norms are smaller because of low prediction error in comparison to learning case, (iv) early residuals are critical for learning and exhibit high gradient norms, (v) residual connections standard deviations significantly impact early residuals gradient norms, (vi) output margin decreases as we remove residual connections from early layers, and (vii) gradient norms strongly correlate with output margins and test-accuracy. We provide the additional results for the same in Sec. E.1, E.5, E.6, E.7, E.8, and E.9.

### E.1 Residual Connections do not impact Memorization but Influences Generalization

In this section, we show that residual connections do not relay memorization but primarily influence generalization, where their removal has no impact on memorization and rather impairs test accuracy. We verify the claim against the following additional models, GPT2-Medium, Smol-LM, Qwen2, TinyViT, BEiT, and DeiT, as presented in Figs. 10g, 10h; Figs. 10k, 10l; Figs. 10i, 10j; Figs. 10a, 10b; Figs. 10c, 10d, and Figs. 10e, 10f, apart from GPT2-Small and ViT-Base results provided in the main paper in Figs. 2c, 2d & Figs. 2a, 2b.

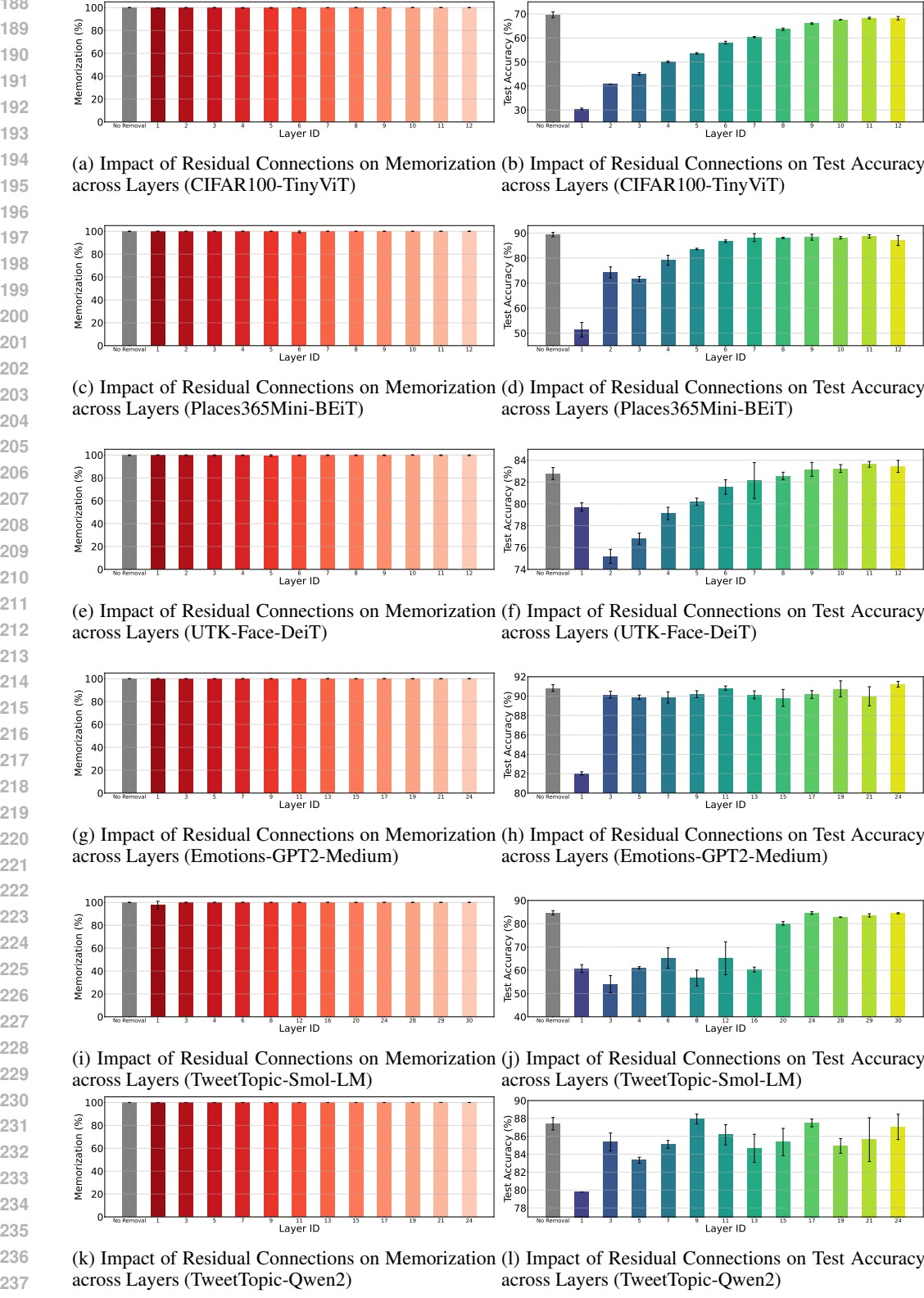

(a) Impact of Residual Connections on Memorization across Layers (CIFAR100-TinyViT)

(b) Impact of Residual Connections on Test Accuracy across Layers (CIFAR100-TinyViT)

(c) Impact of Residual Connections on Memorization across Layers (Places365Mini-BEiT)

(d) Impact of Residual Connections on Test Accuracy across Layers (Places365Mini-BEiT)

(e) Impact of Residual Connections on Memorization across Layers (UTK-Face-DeiT)

(f) Impact of Residual Connections on Test Accuracy across Layers (UTK-Face-DeiT)

(g) Impact of Residual Connections on Memorization across Layers (Emotions-GPT2-Medium)

(h) Impact of Residual Connections on Test Accuracy across Layers (Emotions-GPT2-Medium)

(i) Impact of Residual Connections on Memorization across Layers (TweetTopic-Smol-LM)

(j) Impact of Residual Connections on Test Accuracy across Layers (TweetTopic-Smol-LM)

(k) Impact of Residual Connections on Memorization across Layers (TweetTopic-Qwen2)

(l) Impact of Residual Connections on Test Accuracy across Layers (TweetTopic-Qwen2)

Figure 10: **Residual connections do not influence memorization, but early residuals are critical for learning.** (a) shows that residual connections across all layers have almost no impact on memorization, while (b) highlights that early layers residuals significantly influence test accuracy, indicating their importance for learning, than later ones.

## E.2 ANALYSIS ACROSS HIGHER LABEL NOISE RATIOS

We validate consistency of our claims, i.e., (1) residual connections do not propagate memorization but relay generalization, and (2) early residuals have the most influence on learning, for higher label noise ratios: 5%, 10%, and 20% for Smol-LM and DeiT models across Figs. 11a,11b; Figs. 11c,11d; Figs. 3a,3b; Figs. 11e,11f; Figs. 11g,11h; and Figs. 11i,11j.

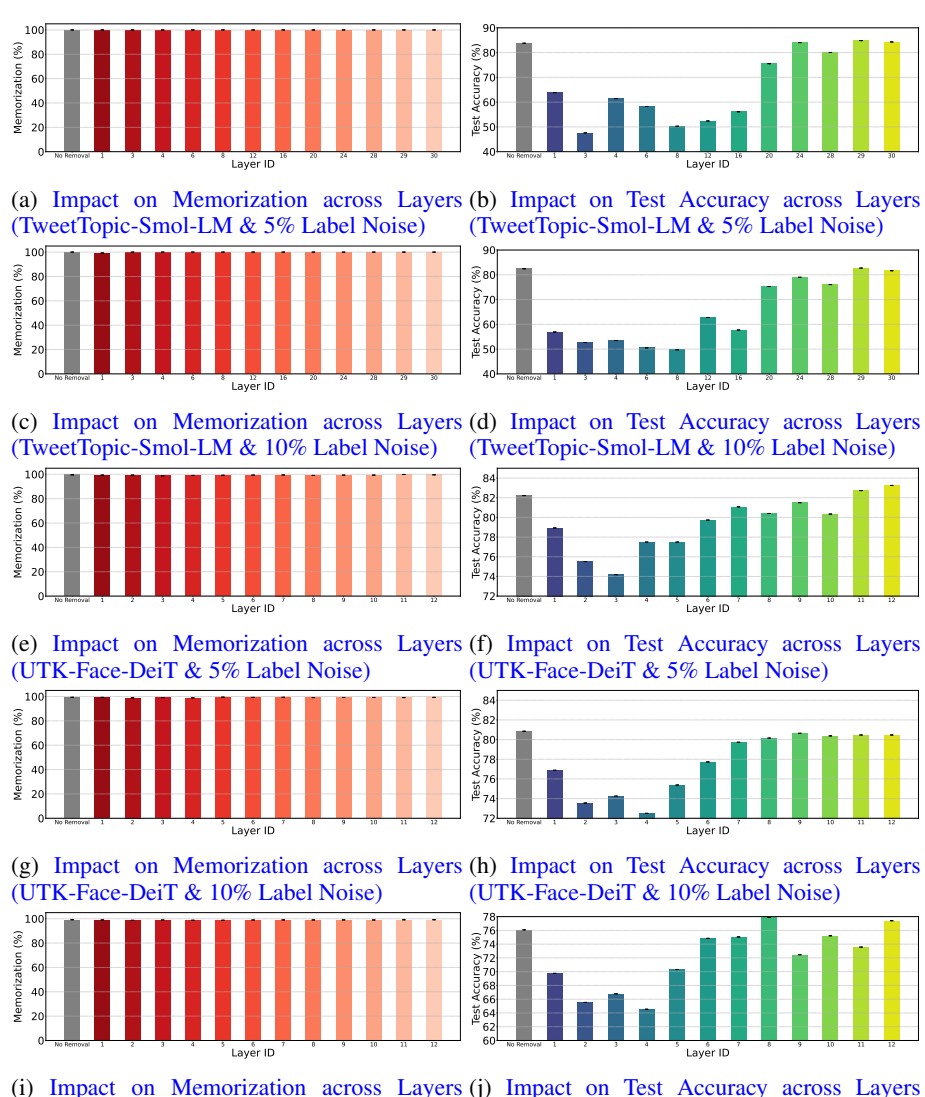

(a) Impact on Memorization across Layers (TweetTopic-Smol-LM & 5% Label Noise)

(b) Impact on Test Accuracy across Layers (TweetTopic-Smol-LM & 5% Label Noise)

(c) Impact on Memorization across Layers (TweetTopic-Smol-LM & 10% Label Noise)

(d) Impact on Test Accuracy across Layers (TweetTopic-Smol-LM & 10% Label Noise)

(e) Impact on Memorization across Layers (UTK-Face-DeiT & 5% Label Noise)

(f) Impact on Test Accuracy across Layers (UTK-Face-DeiT & 5% Label Noise)

(g) Impact on Memorization across Layers (UTK-Face-DeiT & 10% Label Noise)

(h) Impact on Test Accuracy across Layers (UTK-Face-DeiT & 10% Label Noise)

(i) Impact on Memorization across Layers (UTK-Face-DeiT & 20% Label Noise)

(j) Impact on Test Accuracy across Layers (UTK-Face-DeiT & 20% Label Noise)

Figure 11: Consistent results across higher noise ratios. Residual connections do not influence memorization but only relay generalization even for higher noise ratios of 5%, 10%, and 20%. Furthermore, early residuals are the most impactful towards generalization.

### E.3 RESULTS IN GENERATIVE TASKS

To verify the applicability of our claims beyond classification tasks, we carry out the analysis on a generative language modeling task. From Figs. 4a, 4b; Figs. 12a, 12b, we can clearly observe that even in a generative task, the claims that (1) residual connections do not propagate memorization but only relays generalization, and (2) early residuals are the most impactful towards generalization, hold true.

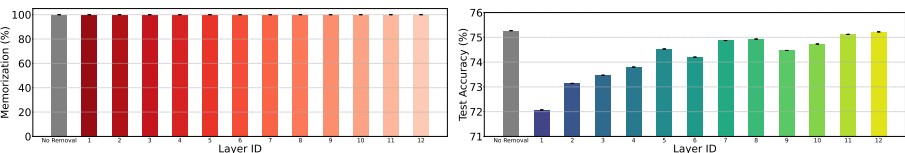

(a) Impact on Memorization across Layers (TweetTopic-GPT2-Small & Generative Task)

(b) Impact on Test Accuracy across Layers (TweetTopic-GPT2-Small & Generative Task)

Figure 12: Results in generative tasks. Consistent with classification tasks, residual connections do not propagate memorization even for generative language modeling tasks, but only relay generalization.

### E.4 SCALING RESIDUAL CONNECTIONS BY FACTOR $c$

We provide another ablation study where instead of completely removing the residual connection, we multiply it by a scaling factor $c$, where $c = [0, 0.25, 0.5, 0.75, 1]$, and $c = 0$ means complete removal and $c = 1$ means no removal, with other values depicting partial removal. We do this analysis for the residual connections in the first transformer layer as they are the most influential layers, and present their influence on memorization and generalization in Figs. 13a & 13b.

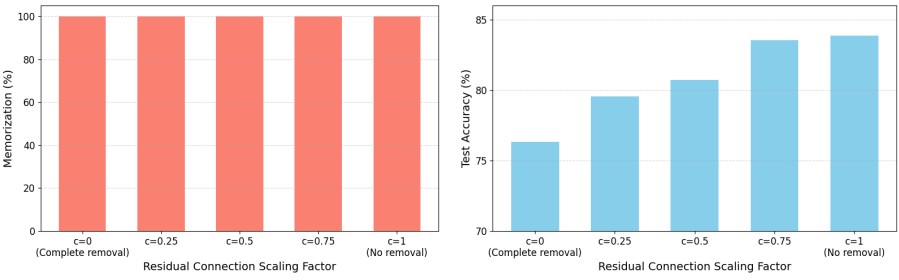

(a) Impact of scaling factor $c$ on memorization (UTK-Face-DeiT)

(b) Impact of scaling factor $c$ on generalization (UTK-Face-DeiT)

Figure 13: Impact of residual connections on memorization and generalization in generative tasks.

From Figs. 13a & 13b, we can clearly observe that even partially removed residual connections can not influence memorization. Apart from that as $c$ decreases from 1 to 0, we can observe a drop in generalization, as we are gradually removing the residual connection, with memorization still remaining intact, which thoroughly aligns with all other our claims made in this paper.

### E.5 MEMORIZATION AND LEARNING ERROR

In this section, we provide the empirical results for confirming that the memorization error is smaller than learning error which causes memorization gradient norms to be smaller than learning gradient norms, as discussed in Section 4.6.

Accordingly, the results are verified against remaining 7 models, GPT2-Small, GPT2-Medium, Smol-LM, Qwen2, ViT-Base, BEiT, and DeiT, as shown in Figs. 14a, 14b, 14c, 14d, 14e, 14f, and 14g, respectively, with TinyViT results present in the main paper in Fig. 7.

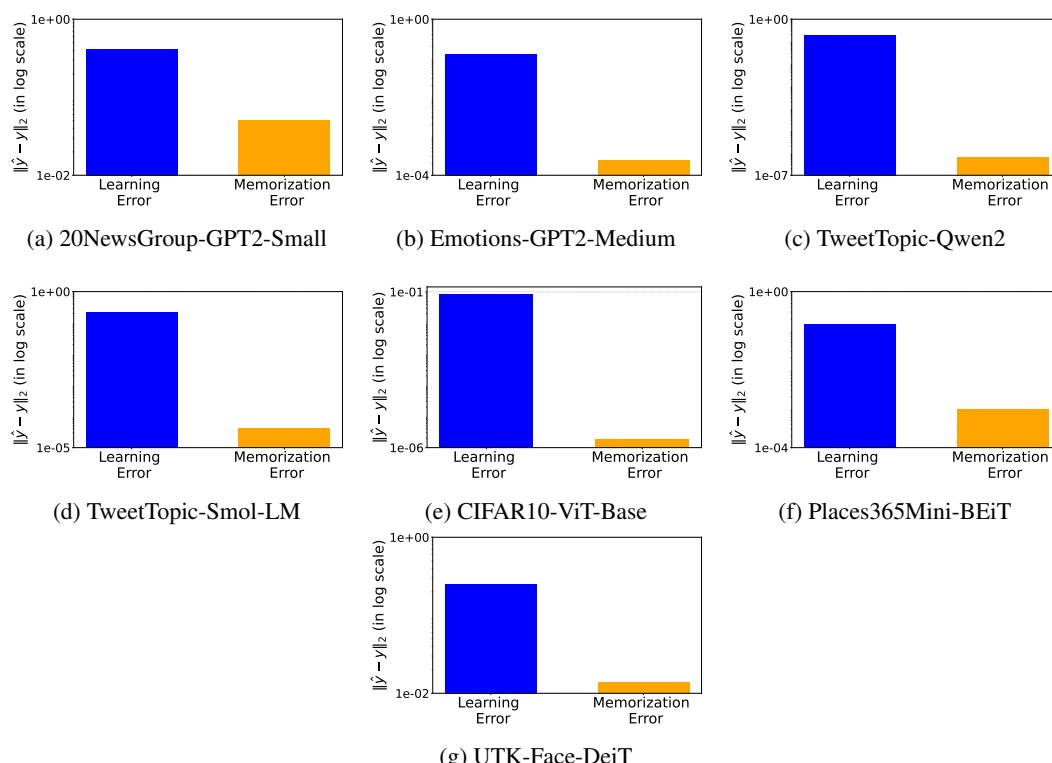

Figure 14: Comparison of memorization and learning errors measured as $\|\hat{y} - y\|_2$ in log scale across datasets and models.

### E.6 GRADIENT NORMS ANALYSIS ACROSS LAYERS

In this section, we provide additional results exhibiting that memorization gradient norm $\|g^{\text{mem}}\|_2$ is smaller than learning gradient norm $\|g^{\text{learn}}\|_2$ across all layers. This explains why residual connections do not impact memorization and only influences learning.

We below present the results for additional models, GPT2-Small, Smol-LM, Qwen2, ViT-Base, BEiT, and DeiT, in Figs. 15a, 15b, 15c, 15d, 15e, and 15f, respectively, apart from GPT2-Medium and TinyViT results which are already shown in the main paper in Figs. 5b & 5a.

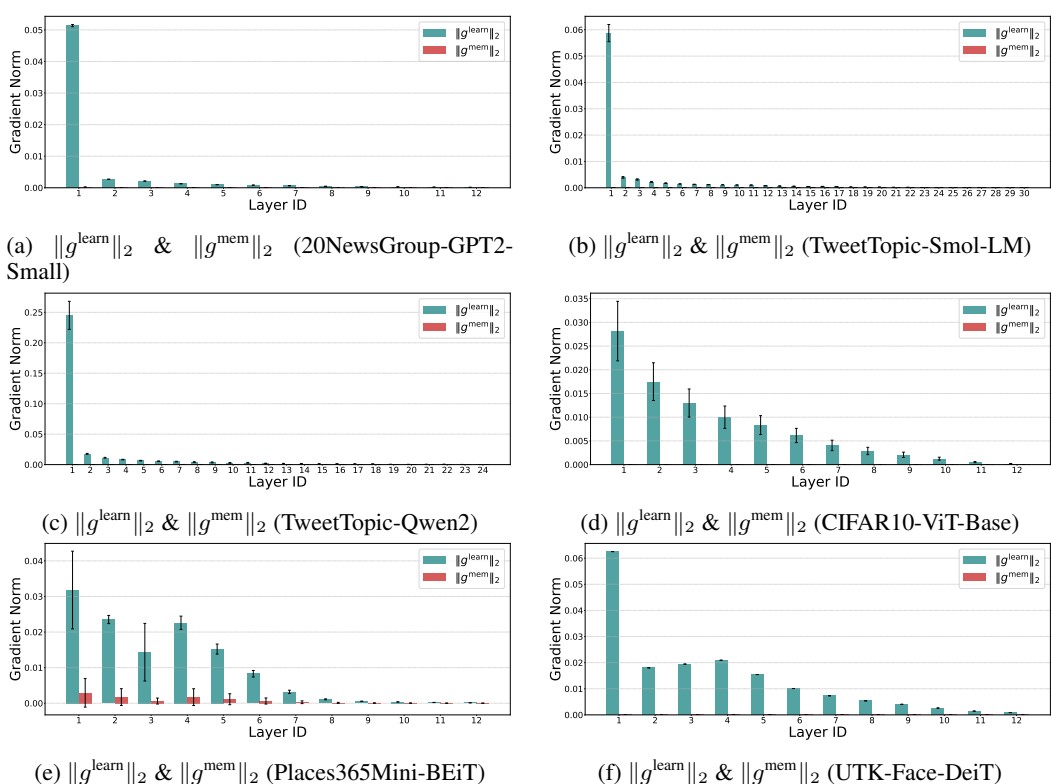

(a) $\|g^{\text{learn}}\|_2$ & $\|g^{\text{mem}}\|_2$ (20NewsGroup-GPT2-Small)

(b) $\|g^{\text{learn}}\|_2$ & $\|g^{\text{mem}}\|_2$ (TweetTopic-Smol-LM)

(c) $\|g^{\text{learn}}\|_2$ & $\|g^{\text{mem}}\|_2$ (TweetTopic-Qwen2)

(d) $\|g^{\text{learn}}\|_2$ & $\|g^{\text{mem}}\|_2$ (CIFAR10-ViT-Base)

(e) $\|g^{\text{learn}}\|_2$ & $\|g^{\text{mem}}\|_2$ (Places365Mini-BEiT)

(f) $\|g^{\text{learn}}\|_2$ & $\|g^{\text{mem}}\|_2$ (UTK-Face-DeiT)

Figure 15: **Memorization gradient norms are consistently smaller than learning gradient norms, with early residuals exhibiting the highest activity.** Across all datasets, $\|g^{\text{mem}}\|_2$ remains significantly lower than $\|g^{\text{learn}}\|_2$ across all layers, explaining why residuals do not influence memorization. The learning gradients peak in early layers, underscoring their critical role in learning.

## E.7 STANDARD DEVIATION ANALYSIS ACROSS LAYERS

In this section, we further show (i) how for both memorization and learning, $\sigma_{W_{j,O}}$, $\sigma_{W_{j,1}}$, $\sigma_{W_{j,2}}$, $\sigma_{x_j}$ & $\sigma_{\tilde{x}_j}$ are of similar magnitudes, and (ii) $\sigma_{x_j}$ & $\sigma_{\tilde{x}_j}$ exhibiting a high variation across layers while rest of the statistics having very less variation.

We present the results for additional models, GPT2-Small, GPT2-Medium, Smol-LM, Qwen2, ViT-Base, BEiT, and DeiT, in Figs. 16, 17, 18, 19,20, 21, and 22, respectively, other than for TinyViT which is already present in the main paper in Fig. 6.

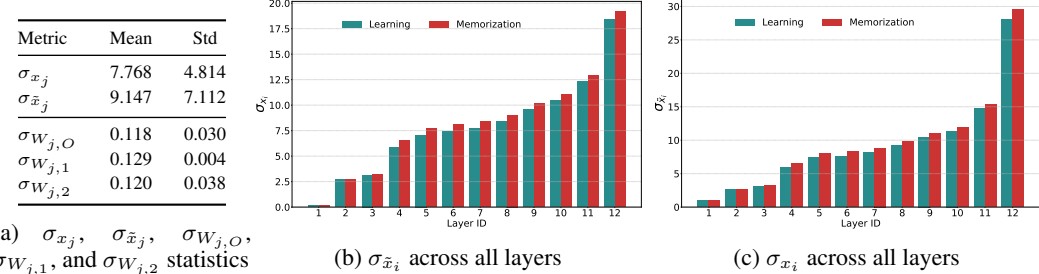

| Metric | Mean | Std |
|--------|------|-----|
| $\sigma_{x_j}$ | 7.768 | 4.814 |
| $\sigma_{\tilde{x}_j}$ | 9.147 | 7.112 |
| $\sigma_{W_{j,O}}$ | 0.118 | 0.030 |
| $\sigma_{W_{j,1}}$ | 0.129 | 0.004 |
| $\sigma_{W_{j,2}}$ | 0.120 | 0.038 |

(a) $\sigma_{x_j}$, $\sigma_{\tilde{x}_j}$, $\sigma_{W_{j,O}}$, $\sigma_{W_{j,1}}$, and $\sigma_{W_{j,2}}$ statistics

(b) $\sigma_{\tilde{x}_i}$ across all layers

(c) $\sigma_{x_i}$ across all layers

Figure 16: **Residual block activations exhibit a high variation across layers but remain consistent between learning and memorization.** The standard deviations of residual connections ($\sigma_{x_j}$, $\sigma_{\tilde{x}_j}$) vary substantially across layers, in contrast to the relatively stable statistics of model parameters ($\sigma_{W_{j,O}}$, $\sigma_{W_{j,1}}$, $\sigma_{W_{j,2}}$). Importantly, $\sigma_{x_j}$ & $\sigma_{\tilde{x}_j}$ statistics are nearly identical for learning and memorization samples. (20NewsGroup-GPT2-Small)

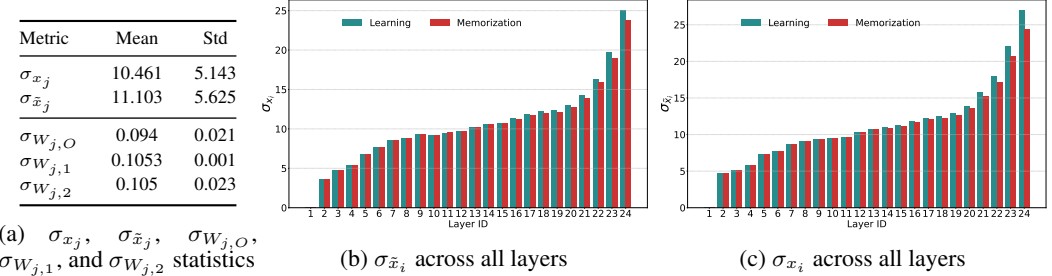

| Metric | Mean | Std |
|--------|------|-----|
| $\sigma_{x_j}$ | 10.461 | 5.143 |
| $\sigma_{\tilde{x}_j}$ | 11.103 | 5.625 |
| $\sigma_{W_{j,O}}$ | 0.094 | 0.021 |
| $\sigma_{W_{j,1}}$ | 0.1053 | 0.001 |
| $\sigma_{W_{j,2}}$ | 0.105 | 0.023 |

(a) $\sigma_{x_j}$, $\sigma_{\tilde{x}_j}$, $\sigma_{W_{j,O}}$, $\sigma_{W_{j,1}}$, and $\sigma_{W_{j,2}}$ statistics

(b) $\sigma_{\tilde{x}_i}$ across all layers

(c) $\sigma_{x_i}$ across all layers

Figure 17: **Residual block activations exhibit a high variation across layers but remain consistent between learning and memorization.** The standard deviations of residual connections ($\sigma_{x_j}$, $\sigma_{\tilde{x}_j}$) vary substantially across layers, in contrast to the relatively stable statistics of model parameters ($\sigma_{W_{j,O}}$, $\sigma_{W_{j,1}}$, $\sigma_{W_{j,2}}$). Importantly, $\sigma_{x_j}$ & $\sigma_{\tilde{x}_j}$ statistics are nearly identical for learning and memorization samples. (Emotions-GPT2-Medium)

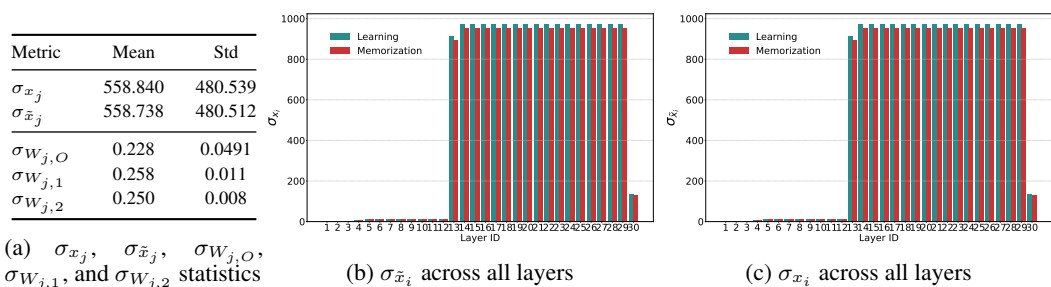

| Metric | Mean | Std |
|---|---|---|
| $\sigma_{x_j}$ | 558.840 | 480.539 |
| $\sigma_{\tilde{x}_j}$ | 558.738 | 480.512 |
| $\sigma_{W_{j,O}}$ | 0.228 | 0.0491 |
| $\sigma_{W_{j,1}}$ | 0.258 | 0.011 |
| $\sigma_{W_{j,2}}$ | 0.250 | 0.008 |

(a) $\sigma_{x_j}$, $\sigma_{\tilde{x}_j}$, $\sigma_{W_{j,O}}$, $\sigma_{W_{j,1}}$, and $\sigma_{W_{j,2}}$ statistics

(b) $\sigma_{\tilde{x}_i}$ across all layers

(c) $\sigma_{x_i}$ across all layers

Figure 18: **Residual block activations exhibit a high variation across layers but remain consistent between learning and memorization.** The standard deviations of residual connections ($\sigma_{x_j}$, $\sigma_{\tilde{x}_j}$) vary substantially across layers, in contrast to the relatively stable statistics of model parameters ($\sigma_{W_{j,O}}$, $\sigma_{W_{j,1}}$, $\sigma_{W_{j,2}}$). Importantly, $\sigma_{x_j}$ & $\sigma_{\tilde{x}_j}$ statistics are nearly identical for learning and memorization samples. (TweetTopic-Smol-LM)

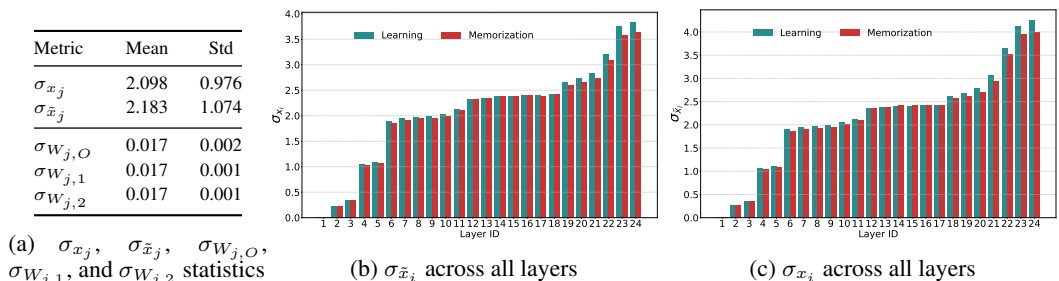

| Metric | Mean | Std |
|---|---|---|
| $\sigma_{x_j}$ | 2.098 | 0.976 |
| $\sigma_{\tilde{x}_j}$ | 2.183 | 1.074 |
| $\sigma_{W_{j,O}}$ | 0.017 | 0.002 |
| $\sigma_{W_{j,1}}$ | 0.017 | 0.001 |
| $\sigma_{W_{j,2}}$ | 0.017 | 0.001 |

(a) $\sigma_{x_j}$, $\sigma_{\tilde{x}_j}$, $\sigma_{W_{j,O}}$, $\sigma_{W_{j,1}}$, and $\sigma_{W_{j,2}}$ statistics

(b) $\sigma_{\tilde{x}_i}$ across all layers

(c) $\sigma_{x_i}$ across all layers

Figure 19: **Residual block activations exhibit a high variation across layers but remain consistent between learning and memorization.** The standard deviations of residual connections ($\sigma_{x_j}$, $\sigma_{\tilde{x}_j}$) vary substantially across layers, in contrast to the relatively stable statistics of model parameters ($\sigma_{W_{j,O}}$, $\sigma_{W_{j,1}}$, $\sigma_{W_{j,2}}$). Importantly, $\sigma_{x_j}$ & $\sigma_{\tilde{x}_j}$ statistics are nearly identical for learning and memorization samples. (TweetTopic-Qwen2)

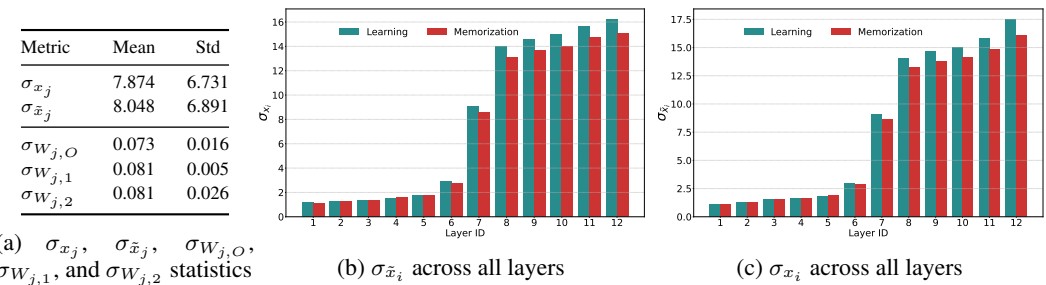

| Metric | Mean | Std |
|---|---|---|
| $\sigma_{x_j}$ | 7.874 | 6.731 |
| $\sigma_{\tilde{x}_j}$ | 8.048 | 6.891 |
| $\sigma_{W_{j,O}}$ | 0.073 | 0.016 |
| $\sigma_{W_{j,1}}$ | 0.081 | 0.005 |
| $\sigma_{W_{j,2}}$ | 0.081 | 0.026 |

(a) $\sigma_{x_j}$, $\sigma_{\tilde{x}_j}$, $\sigma_{W_{j,O}}$, $\sigma_{W_{j,1}}$, and $\sigma_{W_{j,2}}$ statistics

(b) $\sigma_{\tilde{x}_i}$ across all layers

(c) $\sigma_{x_i}$ across all layers

Figure 20: **Residual block activations exhibit a high variation across layers but remain consistent between learning and memorization.** The standard deviations of residual connections ($\sigma_{x_j}$, $\sigma_{\tilde{x}_j}$) vary substantially across layers, in contrast to the relatively stable statistics of model parameters ($\sigma_{W_{j,O}}$, $\sigma_{W_{j,1}}$, $\sigma_{W_{j,2}}$). Importantly, $\sigma_{x_j}$ & $\sigma_{\tilde{x}_j}$ statistics are nearly identical for learning and memorization samples. (CIFAR10-ViT-Base)

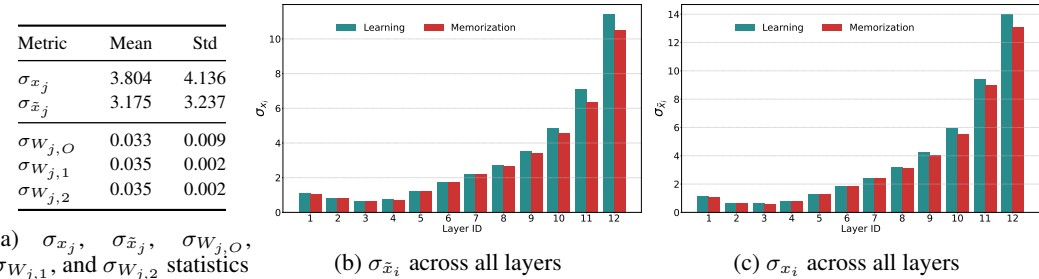

| Metric | Mean | Std |
|---|---|---|
| $\sigma_{x_j}$ | 3.804 | 4.136 |
| $\sigma_{\tilde{x}_j}$ | 3.175 | 3.237 |
| $\sigma_{W_{j,O}}$ | 0.033 | 0.009 |
| $\sigma_{W_{j,1}}$ | 0.035 | 0.002 |
| $\sigma_{W_{j,2}}$ | 0.035 | 0.002 |

(a) $\sigma_{x_j}$, $\sigma_{\tilde{x}_j}$, $\sigma_{W_{j,O}}$, $\sigma_{W_{j,1}}$, and $\sigma_{W_{j,2}}$ statistics

(b) $\sigma_{\tilde{x}_i}$ across all layers

(c) $\sigma_{x_i}$ across all layers

Figure 21: **Residual block activations exhibit a high variation across layers but remain consistent between learning and memorization.** The standard deviations of residual connections ($\sigma_{x_j}$, $\sigma_{\tilde{x}_j}$) vary substantially across layers, in contrast to the relatively stable statistics of model parameters ($\sigma_{W_{j,O}}$, $\sigma_{W_{j,1}}$, $\sigma_{W_{j,2}}$). Importantly, $\sigma_{x_j}$ & $\sigma_{\tilde{x}_j}$ statistics are nearly identical for learning and memorization samples. (Places365Mini-BEiT)

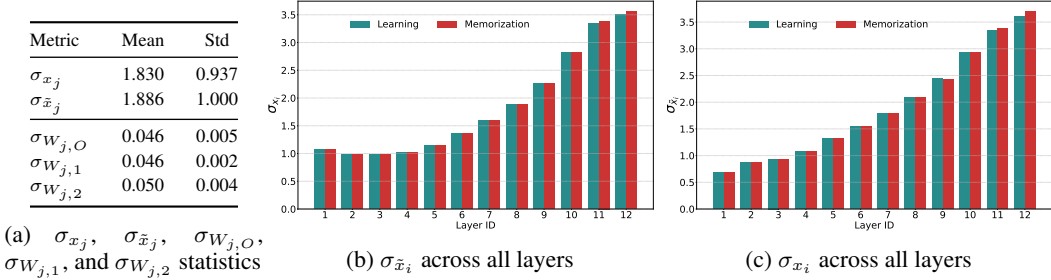

| Metric | Mean | Std |
|---|---|---|
| $\sigma_{x_j}$ | 1.830 | 0.937 |
| $\sigma_{\tilde{x}_j}$ | 1.886 | 1.000 |
| $\sigma_{W_{j,O}}$ | 0.046 | 0.005 |
| $\sigma_{W_{j,1}}$ | 0.046 | 0.002 |
| $\sigma_{W_{j,2}}$ | 0.050 | 0.004 |

(a) $\sigma_{x_j}$, $\sigma_{\tilde{x}_j}$, $\sigma_{W_{j,O}}$, $\sigma_{W_{j,1}}$, and $\sigma_{W_{j,2}}$ statistics

(b) $\sigma_{\tilde{x}_i}$ across all layers

(c) $\sigma_{x_i}$ across all layers

Figure 22: **Residual block activations exhibit a high variation across layers but remain consistent between learning and memorization.** The standard deviations of residual connections ($\sigma_{x_j}$, $\sigma_{\tilde{x}_j}$) vary substantially across layers, in contrast to the relatively stable statistics of model parameters ($\sigma_{W_{j,O}}$, $\sigma_{W_{j,1}}$, $\sigma_{W_{j,2}}$). Importantly, $\sigma_{x_j}$ & $\sigma_{\tilde{x}_j}$ statistics are nearly identical for learning and memorization samples. (UTK-Face-DeiT)

### E.7.1 STANDARD DEVIATION ANALYSIS FOR $C_{\text{attn}}^j$

From Theorem 1, we know that the upper bound of the gradient also depends on $C_{\text{attn}}^j$, across the layers. From Takase et al. (2023), we know that $C_{\text{attn}}^j = 2d_1 h \left( \left( \sqrt{L} + 2 + \frac{1}{\sqrt{L}} \right) \sigma_{Q,j}^3 \sqrt{d_1^3 d_{\text{head}}} + \sigma_{Q,j} \left( \sqrt{d_1} + \sqrt{d_{\text{head}}} \right) \right)$.

We also know that across all transformer layers, $d_1, h, d_{\text{head}}$ would remain the same. Furthermore, due to tokenization and truncation/padding, the input length sequence $L$ is also restricted to a constant value (generally 512 in most of the transformer models). Hence, $C_{\text{attn}}^j$ varies across the layers primarily due to the attention query matrix's standard deviation $\sigma_{Q,j}$. Hence, we check how it varies by computing the standard-deviation of $\sigma_{Q,j}$ for all 8 models, GPT2-Small, GPT2-Medium, Smol-LM, Qwen2, ViT-Base, TinyViT, BEiT, and DeiT, as shown in Table 2.

| Model | Mean | Std |
|---|---|---|
| GPT2-Small | 0.139 | 0.021 |
| GPT2-Medium | 0.111 | 0.013 |
| Smol-LM | 0.247 | 0.0331 |
| Qwen2 | 0.022 | 0.006 |
| ViT-Base | 0.0859 | 0.015 |
| TinyViT | 0.065 | 0.005 |
| BEiT | 0.038 | 0.005 |
| DeiT | 0.046 | 0.004 |

Table 2: Mean and Standard Deviation of $\sigma_{Q,j}$ for all 8 models

From Table 2, we can clearly observe that $\sigma_{Q,j}$ has a very low variance, which means that it does not vary much between layers, for all models.

Hence, in conclusion, it is the residual connections standard-deviations, $\sigma_{x_i}, \sigma_{\tilde{x}_i}$, which primarily influence early layers to have significantly high gradient norms than later layers.

### E.8 OUTPUT MARGINS ANALYSIS ACROSS LAYERS

In this section, we show the importance of early residuals where their removal impacts the model's predictions by making it less confident and hence more prone to misclassifications, and thereby leading to smaller output margins in comparison to later residuals.

We provide the results for remaining models, GPT2-Small, Smol-LM, Qwen2, ViT-Base, TinyViT, BEiT, and DeiT, in Figs. 23a, 23b, 23c, 23d, 23e, and 23f, respectively, other than GPT2-Medium and TinyViT results which are presented in the main paper in Figs. 8b, 8a.

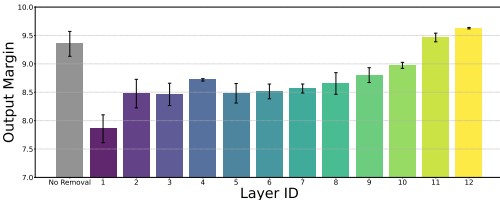

(a) Output Margins after removing residual connections across different layers. (20NewsGroup-GPT2-Small)

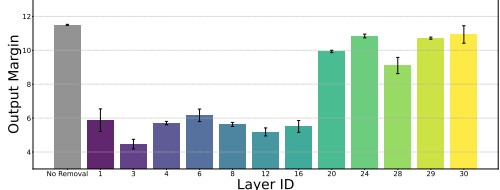

(b) Output Margins after removing residual connections across different layers. (TweetTopic-Smol-LM)

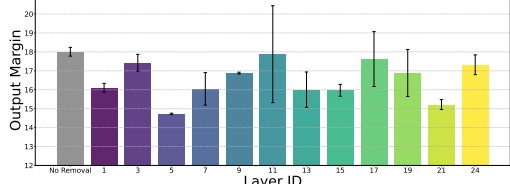

(c) Output Margins after removing residual connections across different layers. (TweetTopic-Qwen2)

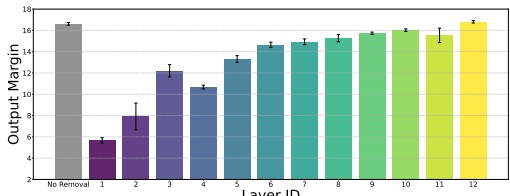

(d) Output Margins after removing residual connections across different layers. (CIFAR10-ViT-Base)

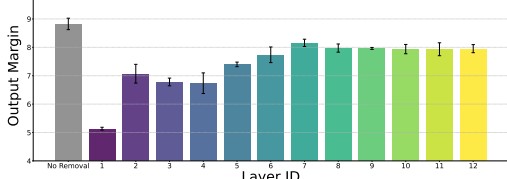

(e) Output Margins after removing residual connections across different layers. (Places365Mini-BEiT)

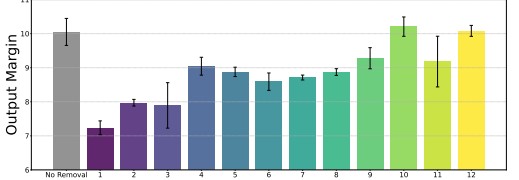

(f) Output Margins after removing residual connections across different layers. (UTK-Face-DeiT)

Figure 23: **Output margin corroborates the importance of early residuals.** Removing residual connections from early layers drastically reduces the output margin, increasing uncertainty and misclassifications. In contrast, removing later residuals has a smaller effect—highlighting that early residuals play a crucial role in enabling confident learning.

### E.9 UNIFIED VIEW OF GRADIENT NORM, OUTPUT MARGIN AND TEST ACCURACY

In this section, we further strengthen the observation that residuals (early residuals) which exhibit higher gradient norms, when removed, leads to smaller output margins because the model's prediction confidence decreases, and hence it leads to more misclassifications and high drop in test accuracy, in comparison to remove residuals with smaller gradient norms (later residuals) whose removal has discernible impact on the output margin and test accuracy.

We provide the results for all the models, GPT2-Medium, Smol-LM, Qwen2, TinyViT, BEiT, and DeiT, in Figs. 24a, 24b, 24c, 24d, 24e, and 24f, respectively, apart from GPT2-Small and ViT-Base results that are presented in the main paper in Figs. 9b, 9a.

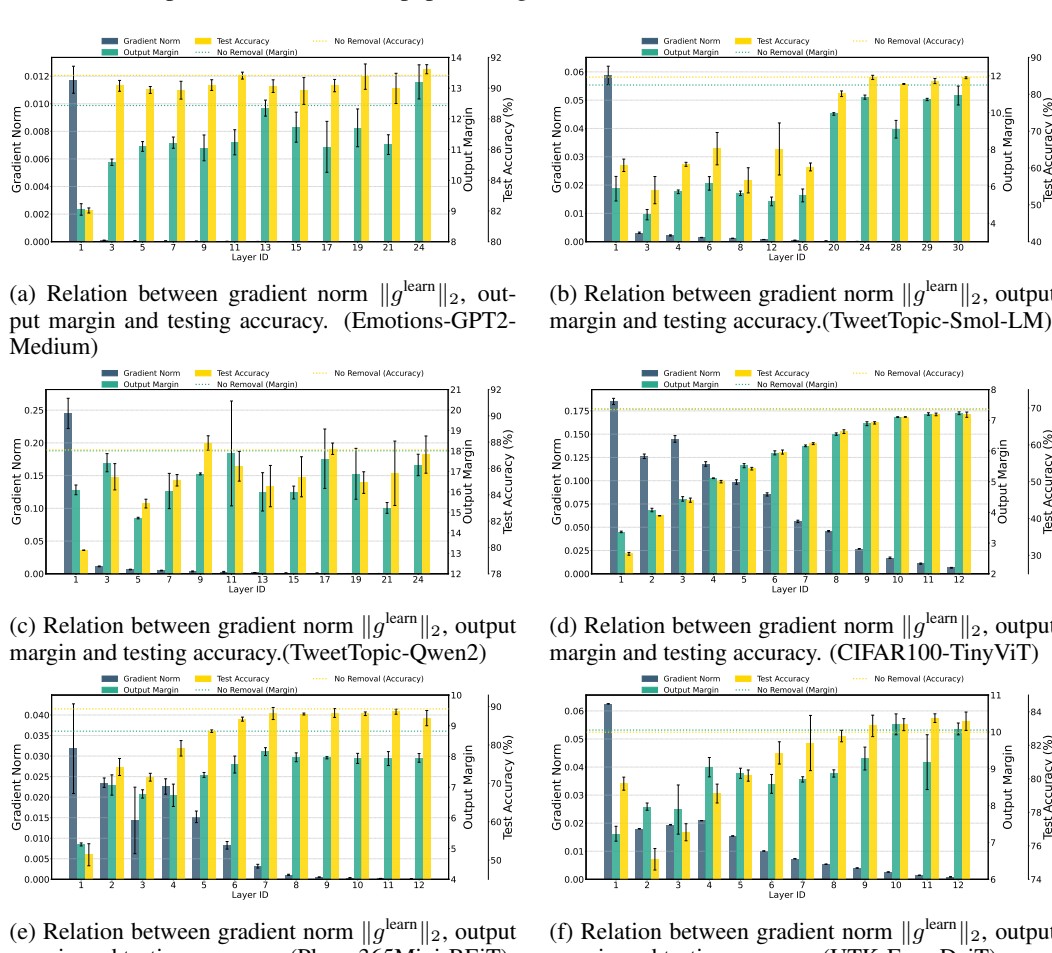

(a) Relation between gradient norm $\|g^{\text{learn}}\|_2$, output margin and testing accuracy. (Emotions-GPT2-Medium)

(b) Relation between gradient norm $\|g^{\text{learn}}\|_2$, output margin and testing accuracy.(TweetTopic-Smol-LM)

(c) Relation between gradient norm $\|g^{\text{learn}}\|_2$, output margin and testing accuracy.(TweetTopic-Qwen2)

(d) Relation between gradient norm $\|g^{\text{learn}}\|_2$, output margin and testing accuracy. (CIFAR100-TinyViT)

(e) Relation between gradient norm $\|g^{\text{learn}}\|_2$, output margin and testing accuracy. (Places365Mini-BEiT)

(f) Relation between gradient norm $\|g^{\text{learn}}\|_2$, output margin and testing accuracy.(UTK-Face-DeiT)

Figure 24: **Gradient norm correlates with output margin and test accuracy.** Residual connections with higher $\|g^{\text{learn}}\|_2$ (early residuals) yield lower output margins and degraded accuracy when removed, indicating their greater importance for learning. Conversely, residuals with lower $\|g^{\text{learn}}\|_2$ (later residuals) have minimal effect on margin and accuracy when removed.

### E.10 GRADIENT NORMS AND OUTPUT MARGIN ANALYSIS ACROSS EPOCHS

We provide further analyses into the output margin and gradient norms over the course of training, to understand how they evolve.

#### E.10.1 GRADIENT NORM ANALYSIS OVER EPOCHS

We study how the gradient norm evolves over the course of training when the residual connections are present and absent. We specifically focus on removing the residual connections in the first layer, as in general, they are the most impactful. For the gradient norms, we simply track the gradient norms for the early (1st layer), middle (6th layer) and the later layer (12th layer) for DeiT model, to get a general sense of how the gradient norms evolve across different layers.

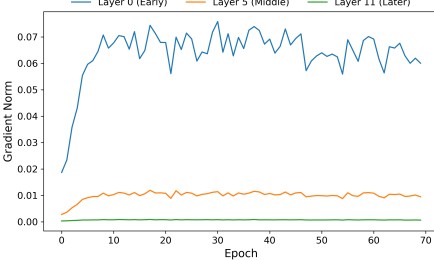 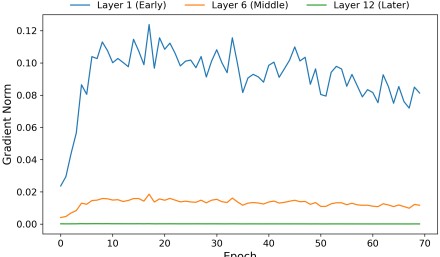

(a) Gradient norm analysis across layers (UTK-Face & DeiT with no removal)  (b) Gradient norm analysis across layers (UTK-Face & DeiT, removing layer 1 residual connections)

Figure 25: Gradient norms evolve gradually over training, with early layers exhibiting higher gradient norms in comparison to middle/later layers.

From Figs. 25a & 25b, we can clearly understand that the gradient norms evolve gradually over epochs with early layers exhibiting much higher gradient norms in comparison to middle and later layers over the course of training.

#### E.10.2 OUTPUT MARGIN ANALYSIS OVER EPOCHS

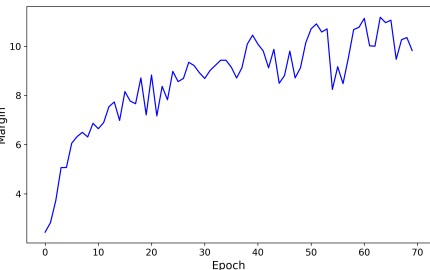 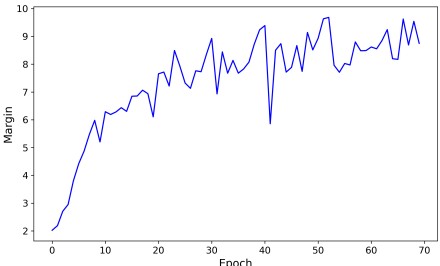

(a) Output margin analysis over epochs (UTK-Face & DeiT with no removal)  (b) Output margin analysis over epochs (UTK-Face & DeiT, removing layer 1 residual connections)

Figure 26: Output margins evolve gradually over epochs and output margins after residual connections removal are smaller than without removal, since generalization is impacted.

We study how the output margins evolve over the course of training when the residual connections are present and absent. We specifically focus on removing the residual connections in the first layer

as in general, they are the most impactful. From Figs. 26a & 26b, we can observe that similar to gradient norms, output margins evolve gradually across training. Furthermore, the output margins corresponding to removing residual connections are generally smaller than without any removal over the course of training. This explains that generalization is impacted when the residual connections are removed.

