# OpenReview forum: "Residual Connections Relay Generalization but Not Memorization in Transformers"
_ICLR.cc/2026/Conference — Submitted to ICLR 2026_

### Official Review · Reviewer_Q3pD · 2025-10-23

**Soundness:** 2
**Presentation:** 2
**Contribution:** 1
**Rating:** 2
**Confidence:** 4

**Summary:**

It studies the role of residual connection on memorization and generalization in transformers, finding that removing one residual connection does not affect memorization, but affect generalization when the removal happens in early layers.

**Strengths:**

It does detailed ablation experiments, and train models on multiple datasets.

**Weaknesses:**

It's not surprising that deeper layers are less effective e.g. see [1]. It's expected that removing residual connection at deeper layers has less effects on the test accuracy, and early residuals are critical for learning is not something new.

If we can stably train LLMs with residual connection removed, it's expected that it can memorize the training data since it's overparameterized.

The noise label rate is fixed to 1% for all experiments, the claims may no longer hold when noise label rate becomes 10%.

[1] The Curse of Depth in Large Language Models

**Questions:**

1. What are the gradient norm dynamics during training for the models with residual connection removed? It would be better if they are provided along with the theoretical analysis of gradient norms
2. In each decoder layer, there are two residual connections, do you remove both of them or just one of them? After removal, is the training still stable?

---

> ### Author Response · Authors · 2025-11-18
> **Response to reviewer Q3pD (1/2)**
>
> **We thank the reviewer for taking the time to review our paper. Below, we address the reviewer’s questions in detail, while providing the requested additional experiments with our best effort.**
>
> >**Weaknesses:**
> >
> >**W1 and W2:**
> >
> >It's not surprising that deeper layers are less effective e.g. see [1]. It's expected that removing residual connection at deeper layers has less effects on the test accuracy, and early residuals are critical for learning is not something new. If we can stably train LLMs with residual connection removed, it's expected that it can memorize the training data since it's overparameterized.
> >
> >**Our Response:**
> >
> > Thank you for your comment. We believe the reviewer’s concern stems from a misunderstanding of our motivation and experimental setup. Therefore, please let us try to  re-iterate and make clearer our core contributions and motivation:
>
> >As explained in the Introduction, our scope of interest is dedicated to residual connections. We were genuinely wondering whether residual connections can propagate memorization, primarily because they are responsible for information propagation from one layer to the other, while bypassing the intermediate transformations. This information could even contain **memorized signals** alongwith the **generalization signals**, even though residual connections do not consist of any model parameters. Furthermore, there is no existing work that studied their impact on memorization, making our work cleanly **novel**. Importantly, we would like to highlight that our finding that the model continues to memorizes 100% of the noisy samples  even after removing residual connections is not an obvious tautological outcome. This is primarily because as stated in our motivation, residual connections  are known to carry information, which could have plausibly included memorized information. Hence understanding whether their removal affects memorization is therefore a meaningful and nontrivial question, which is not explored in prior work. We genuinely believe someone in the research community must answer this question, as no existing work has done it. As models grow more complex, understanding which components carry and do not carry memorized information is crucial for model interpretability and improved model design.
>
> >Coming to the distinction between our work and [1]. There is an important difference in [1] and ours which have clearly stated in Section 5.1. The experiments done in [1] are focused on removing entire transformer layers which remain their study at a coarse-grained level. However, in our paper we pay attention specifically on the **residual connection**. Furthermore, [1] does not study the influence of residual connections on memorization which marks a clear distinction between ours and their work. Lastly, we have even supported our early residual connections influence experiment through a gradient analysis experiment (Fig 3, Theorem 1 and Theorem 2) followed by the standard-deviation analysis experiment (Fig 4) of the residual connections, which has not been done in [1] and neither in any other prior work. We hope this clarification helps resolve the reviewer’s confusion regarding the core novel  contributions of our work.
>
>
> >**W3:** The noise label rate is fixed to 1% for all experiments, the claims may no longer hold when noise label rate becomes 10%.
> >
> >**Our Response:**
> >
> >We take into account the reviewer’s suggestions and carry out our analysis even on higher label noise ratios of 5%, 10%, and 20% and present the results in **Section 4.3**. Clearly, from the results above, we can see that our main claims remain valid even when the noise ratios are increased, i.e., (1) residual connections only propagate generalization but not memorization, and (2) early layers residual connections are the most impactful.

---

> > ### Author Response · Authors · 2025-11-18
> > **Response to reviewer Q3pD (2/2)**
> >
> > >**Questions:**
> > >
> > >**Q1:** What are the gradient norm dynamics during training for the models with residual connection removed? It would be better if they are provided along with the theoretical analysis of gradient norms
> > >
> > >**Our Response:**
> > >
> > >Following the reviewer’s suggestion, we provide a comparative analysis of the gradient norm analysis during training for models with and without residual connections and present the results in **Appendix E.10**.
> > From the results, we can clearly see that the gradient norms evolve gradually across training in both cases, and early layers exhibit higher gradient norms in comparison to middle and later layers..
> >
> >
> > >**Q2:** In each decoder layer, there are two residual connections, do you remove both of them or just one of them? After removal, is the training still stable?
> > >
> > >**Our Response:**
> > >
> > >In our layer-wise ablation analysis, we remove both residual connections from the selected transformer layer simultaneously. We have added this point in our revised version in **line 161**, thanks to your comment. The model achieves 100% train accuracy even after removing residual connections.
> >
> >
> > **We hope our response resolves your concerns and questions, and helps you consider raising your score. If there is anything that we can further clarify, please let us know. We appreciate your time and effort again.**
> >
> > [1] Sun, Wenfang, et al. "The curse of depth in large language models.", NeurIPS 2025

---

> > ### Comment · Reviewer_Q3pD · 2025-11-20
> >
> > 1. "residual connections are known to carry information, which could have plausibly included memorized information. Hence understanding whether their removal affects memorization is therefore a meaningful and nontrivial question"
> >
> > "carry information" and "memorization" are not necessarily related. What does "carry information" mean here? Any relevant works? Residual connection is simply $x+f(x)$, people use it only because it can stabilize the gradient flow during training. If a model can be stably trained without residual connection, no one would use residual connection.
> > This operation should not affect the expressiveness of the model. For most tasks (except reconstruction task, where the label $y$ is the same as input $x$), I don't see any intuition why $x+f(x)$ memorize more than $f(x)$.
> >
> > 2. "the model continues to memorizes 100% of the noisy samples even after removing residual connections is not an obvious."
> >
> > Why it's not obvious? The number of parameters does not change after removing residual connection. It seems very expected to me.

---

> > > ### Author Response · Authors · 2025-11-20
> > >
> > > **We thank the reviewer very much for promptly getting back to us. Below we address their follow-up questions:**
> > >
> > > ---
> > >
> > > **Q1 Response:**
> > >
> > > Yes, residual connections are utilized for stable gradient flow. However, that does not necessarily mean residual connections convey no memorization. Prior studies [1,2,3] support this by showing that residual connections can have harmful impacts. This hints that the “information” transferred by residual connections may also contain negative, harmful signals. Since memorization hurts generalization and hence is harmful, based on prior work, it is fair to ask whether “residual connections can propagate memorization or not”. By the way, we must correct a point of yours: residual connection is `x`, not `x+f(x)`.
> > >
> > > We hope this clarifies the reviewer’s question, and we are happy to address any further concerns or questions that the reviewer may have.
> > >
> > >
> > > ---
> > >
> > > **Q2 Response:**
> > >
> > > We respectfully disagree with the reviewer. We should reiterate our previous response, and we do not believe it is fair to make such an assumption in the first place that only parameterized layers memorize data. Because if having no parameters conveys no memorization, one might argue that having no parameters may convey no generalization signals either. However, we know that even though residual connections do not have parameters, they transfer signals, `x`, which can contain harmful signals [1,2,3]. Hence, our motivating question was whether the signals contain either or both memorization and generalization, and this paper thoroughly examined it.
> > >
> > > We hope our response resolves your question. If there is anything that we can further clarify, please let us know. We appreciate your time and effort again.
> > >
> > >
> > > ---
> > >
> > > \
> > > [1] Wu, Dongxian, et al. "Skip connections matter: On the transferability of adversarial examples generated with resnets.", ICLR 2020
> > >
> > > [2] Hao, Koh Jun, et al. "On the Vulnerability of Skip Connections to Model Inversion Attacks.", ECCV 2024
> > >
> > > [3] Kelesis, Dimitrios, Dimitris Fotakis, and Georgios Paliouras. "Analyzing the effect of residual connections to oversmoothing in graph neural networks." Machine Learning 2025

---

> > > > ### Author Response · Authors · 2025-11-24
> > > > **Gentle reminder to review the rebuttal**
> > > >
> > > > We would greatly appreciate it if the reviewer could review our rebuttal and indicate whether our responses resolve their concerns. We are happy to provide additional explanations or extend the discussion if needed.

---

> > > > ### Comment · Reviewer_Q3pD · 2025-11-28
> > > >
> > > > I still find it intuitive that removing residual connection slightly hurts generalization and does not affect memorization, since memorization is much easier than generalization.
> > > >
> > > > You only remove residual connection from one layer, so the expressivity of models before and after removal should be roughly the same. The model after removing one layer's residual connection is slightly less expressive and the training is less stable, thus get worse on generalization, but it's still easy to memorize all label flipped data.

---

> > > > > ### Author Response · Authors · 2025-12-01
> > > > >
> > > > > We thank the reviewer for continuing to engage with our rebuttal. We would like to clarify that our work does not assume a priori that residual connections cannot propagate memorization. Regarding the reviewer’s point that memorization is “much easier” than generalization: we are not aware of any empirical or theoretical evidence establishing this claim in the context of deep networks, and thus it should not be taken as an assumption without justification. Nevertheless, even if it were true, our one-layer ablation study remains sound. That is because, if memorization were indeed easier than generalization, then removing a residual connection from one layer, in principle, could have impacted memorization but not affected generalization. However, our findings clearly show that this ablation consistently unchanges memorization while hurting generalization. This asymmetric effect is exactly what our paper highlights. We hope this resolves the reviewer’s concern.

---

### Official Review · Reviewer_Lgg1 · 2025-10-27

**Soundness:** 3
**Presentation:** 3
**Contribution:** 2
**Rating:** 6
**Confidence:** 4

**Summary:**

This paper proposes an empirical investigation exploring the impact of residual connections on generalization and memorization in transformer architectures. The authors perform a series of experiments using various architectures (GPT2, Small-LM, Qwen2, ViT-B, etc.) and datasets (Emotions, 20Newsgroup, CIFAR, etc.). The study's main findings are:

- Removing residual connections has a negative effect on generalization, but not on memorization.
- Early residual connections help the most with generalization capabilities.
- The norm of the gradient with respect to the residual input is higher for natural examples than for noise examples.

**Strengths:**

- The paper proposes a novel empirical study examining the effect of residual connections on memorization in transformer architectures.
- The authors perform a broad set of experiments, considering 8 architecture variants and 7 datasets.
- Empirical results clearly show the impact of residual connections on generalization (Fig. 1).
- The authors propose an analysis to explain why residual connections help with generalization, investigating the effect on the gradient norm.

**Weaknesses:**

- The residual connection is one of the most standard components of modern deep learning architectures. While this study explores a novel aspect of residual connections, it is unclear how one could use these insights to improve existing architectures. So, while the contribution showcases a new effect of residual connections—which is valuable—it is unclear how one could build on this observation.
- Most of the experiments use 1% label noise as the setup. How robust are the paper's findings for several levels of noise (1%, 5%, 10%, 50%, etc.)?
- Similarly, it seems that most of the gradient analyses are done after model training. How do the different metrics (gradient norm, output margin, etc.) evolve during training? Would the conclusions be different at different times during training?
- In line 300, the authors make the assumption that x^C ≈ x^NL as they come from the same class. This seems quite strong and unnecessary, as the authors verify empirically that the standard deviations are similar for the two examples.

**Questions:**

See Weaknesses.

---

> ### Author Response · Authors · 2025-11-18
> **Response to reviewer Lgg1 (1/1)**
>
> **We thank the reviewer for their time in reviewing our paper, while providing constructive feedback. Below, we address the points they raised and provide additional supporting experiments.**
>
> >**Weaknesses:**
> >
> >**W1:** The residual connection is one of the most standard components of modern deep learning architectures. While this study explores a novel aspect of residual connections, it is unclear how one could use these insights to improve existing architectures. So, while the contribution showcases a new effect of residual connections—which is valuable—it is unclear how one could build on this observation.
> >
> >**Our Response:**
> >
> >The goal of our paper is to establish a precise understanding of the transformer architecture by identifying which components do and do not carry memorization. This is particularly important for future efforts aimed at suppressing memorization in transformers. Our analysis demonstrates, for the first time, that residual connections do not propagate memorization; they transmit only generalization signals. Knowing what does not transfer memorization is a necessary step toward isolating the components that do. We believe this work provides a clearer mechanistic understanding of residual connections in terms of memorization and generalization in transformers, laying the groundwork for future research, such as not associating residual connections with memorization and looking into other transformer components to combat memorization. As another example of potential implication, future practitioners can envision upsampling the generalization signals due to residual connections—since they do not transfer memorization—may help to counterbalance memorization due to other components, based on our novel interpretability insights.
>
>
> >**W2:** Most of the experiments use 1% label noise as the setup. How robust are the paper's findings for several levels of noise (1%, 5%, 10%, 50%, etc.)?
> >
> >**Our Response:**
> >
> >We take into account the reviewer’s suggestions and carry out our analysis even on higher label noise ratios of 5%, 10%, and 20% and present the results in **Section 4.3**. Clearly, from the above results we can see that our main claims remain valid even when the noise ratios are increased, i.e., (1) residual connections only propagate generalization but memorization, and (2) early layers residual connections are the most impactful.
>
>
> >**W3:** Similarly, it seems that most of the gradient analyses are done after model training. How do the different metrics (gradient norm, output margin, etc.) evolve during training? Would the conclusions be different at different times during training?
> >
> >**Our Response:**
> >
> >We take into account the reviewer’s thoughts and carry out the analysis of the gradient norms and output margins across training and provide the results in **Appendix E.10**. From the results we can see that both the gradient norms and output margins evolve gradually during the course of training. Furthermore, early layers gradients norms are always higher than middle and later layers. Thereby showing that Theorem 2 of our paper holds true even during the course of training.
>
>
> >**W4:** In line 300, the authors make the assumption that $x^C$ ≈ $x^{NL}$ as they come from the same class. This seems quite strong and unnecessary, as the authors verify empirically that the standard deviations are similar for the two examples.
> >
> >**Our Response:**
> >
> >We considered this assumption because the 2 samples semantically belong to the same class label and hence they tend to share similar features for that class. Hence, it becomes fair to consider that  $x^C$ ≈ $x^{NL}$, which further implies that the intermediate forward pass representations become approximately similar as mentioned in Eq.4 of our paper. We even verified this in Fig.4b and 4c. This approximation enables us to understand how the difference in learning and memorization gradient norms primarily stems from the prediction error but not the intermediate residual connections and model parameters statistics. We hope this clarifies the question.
>
>
> **We hope our response resolves your concerns and questions, and helps you consider raising your score. If there is anything that we can further clarify, please let us know. We appreciate your time and effort again.**

---

> ### Author Response · Authors · 2025-11-24
> **Gentle reminder to review the rebuttal**
>
> We would appreciate it if the reviewer could take a look at our rebuttal and let us know whether it answers their questions. We will be happy to elaborate on any point that may require more detail.

---

### Official Review · Reviewer_o1jb · 2025-10-31

**Soundness:** 2
**Presentation:** 2
**Contribution:** 2
**Rating:** 4
**Confidence:** 3

**Summary:**

This paper demonstrates, both experimentally and theoretically, that residual connections affect generalization performance but do not contribute to memorization on training data with added noise. It compares test error with and without residual connections, and memorization measured on training data with 1% label noise. As a result, it experimentally shows that the presence or absence of residual connections influences test error but does not affect memorization. The paper also provides a theoretical interpretation via upper bounds on gradient norms to explain these experimental results. In particular, it shows that the impact of residual connections is larger in the shallow layers.

**Strengths:**

- By comparing gradient norms in residual connections, this paper shows that the learning gradient norm is larger than the memorization gradient norm, thereby demonstrating, both experimentally and theoretically, that residual connections tend to transmit learning-relevant information through gradients.

- It was experimentally shown that, particularly in the shallow layers, residual connections have a large impact on test performance.

**Weaknesses:**

- Since residual connections have no parameters and do not memorize data, it seems natural that, if the training loss can be driven to zero, removing residual connections would not change memorization performance. In that respect, there appears to be little novelty. Moreover, performance improvements from adding residual connections were demonstrated in the original ResNet paper, so the novelty also seems limited in that sense.

- Parameterized layers memorize data. Given that, it seems that removing residual connections would not change memorization. The argument based on upper bounds of gradient norms is indirect, and an inequality between the learning gradient norm and the memorization gradient norm only states an ordering, which is weak to support the high memorization reported in the experiments.

- Because residual connections pass information forward, removing them in the early layers impedes the flow to the deeper layers of the network, so it is to be expected that learning in those deep layers will be affected. Prior work has described this as learning a coarse-grained view, so this seems to be a restatement. In addition, the inequality presented in Section 5.1 is not actually proved.

**Questions:**

- Rather than removing residual connections entirely, one could multiply them by a suitable coefficient and study their effect continuously. This would make it possible to show whether the phenomena reported in the paper occur only when residual connections are completely removed, or whether they can also arise when the residual pathway is merely weakened. Eliminating residual connections is a large intervention on the model, and there is a possibility that the reported drop in generalization performance simply reflects continuing to use hyperparameters tuned for a model with residual connections.

- Since the parameter count does not change with or without residual connections, might the memorization metric be essentially determined by model size? Note that from a training standpoint, if residual connections are entirely absent, increasing depth can cause vanishing gradients, so the difficulty of memorizing the data may simply increase.

- The current experiments use only 1% noise, which is a restrictive setting. Experiments with other noise levels, such as 5% or 10%, are natural to consider, so what is the reason for using only 1%?

---

> ### Author Response · Authors · 2025-11-18
> **Response to reviewer o1jb (1/2)**
>
> **We thank the reviewer for their thoughtful and constructive feedback. Below, we address their questions in detail, while covering the requested additional experiments.**
>
>
> >**Weaknesses**
> >
> >**W1:** Since residual connections have no parameters and do not memorize data, it seems natural that, if the training loss can be driven to zero, removing residual connections would not change memorization performance. In that respect, there appears to be little novelty. Moreover, performance improvements from adding residual connections were demonstrated in the original ResNet paper, so the novelty also seems limited in that sense.
> >
> >**Our Response:**
> >
> >We respectfully disagree with the reviewer’s point. Because if having no parameters conveys no memorization, one might argue that having no parameters may convey no generalization signals either. However, we verified that the residual connections propagate generalization signals but not memorization signals. This is the whole idea and contribution of this paper, which no one in the community has explored yet. We hope this clean novelty is well conveyed.
>
> >**W2:** Parameterized layers memorize data. Given that, it seems that removing residual connections would not change memorization. The argument based on upper bounds of gradient norms is indirect, and an inequality between the learning gradient norm and the memorization gradient norm only states an ordering, which is weak to support the high memorization reported in the experiments.
> >
> >**Our Response:**
> >
> >We respectfully disagree with the reviewer. We do not believe it is fair to make such an assumption in the first place that only parameterized layers memorize data. Even though residual connections do not have parameters, they transfer signals. Hence, our motivating question was whether the signals contain either or both memorization and generalization, and this paper thoroughly examined it.
>
> >Regarding the gradient analysis experiment, our motivation was to understand why residual connections do not propagate memorization. Therefore, to interpret this behavior we carried out the gradient norm analysis, which reveals that memorization gradient norms are smaller than learning gradient norms across all layers. Therefore, providing a direct indication of why residual connections do not propagate memorization, an argument that has not been studied in prior literature.
>
> >**W3:** Because residual connections pass information forward, removing them in the early layers impedes the flow to the deeper layers of the network, so it is to be expected that learning in those deep layers will be affected. Prior work has described this as learning a coarse-grained view, so this seems to be a restatement. In addition, the inequality presented in Section 5.1 is not actually proved.
> >
> >**Our Response:**
> >
> >The reviewer themself agrees that residual connections pass information forward from one layer to the other, and removing early residuals should “impede” this flow. However, the reviewer has forgotten that the propagated information could be either or both memorization and generalization, which makes our paper’s results novel and interesting, where we show that after removal of residual connections memorization is not impeded and rather only generalization.
>
> >Regarding the inequality in Section 5.1, its purpose is to mathematically summarize the empirical trend observed in Figs. 2b and 2d—namely, that early residual connections have a larger effect on test accuracy. To support this observation theoretically, we conducted a gradient-norm analysis and formally proved in Theorem 2 that early-layer residuals exhibit higher upper bounds on gradient norms than later layers. This provides the intuitive and mathematical rationale behind why early residuals have greater impact.

---

> > ### Author Response · Authors · 2025-11-18
> > **Response to reviewer o1jb (2/2)**
> >
> > >**Questions:**
> > >
> > >**Q1:** Rather than removing residual connections entirely, one could multiply them by a suitable coefficient and study their effect continuously. This would make it possible to show whether the phenomena reported in the paper occur only when residual connections are completely removed, or whether they can also arise when the residual pathway is merely weakened. Eliminating residual connections is a large intervention on the model, and there is a possibility that the reported drop in generalization performance simply reflects continuing to use hyperparameters tuned for a model with residual connections.
> > >
> > >**Our Response:**
> > >
> > >Our central objective is to isolate and understand the influence of residual connections themselves on memorization and generalization in transformers.  Hence, we examined this by fully removing the connections, in order not to allow partial information to pass through the residual pathway, otherwise we would not be able to see a clean impact of residual connection.
> >
> > >That being said, we respect the reviewer’s comment and conducted an additional experiment where we partially attenuate the residual connection in the first transformer layer—the layer with the highest measured influence—by multiplying it with a coefficient c∈{0,0.25,0.5,0.75,1}. Please refer to the results in **Appendix E.4.** In summary, across all intermediate scaling values c∈{0.25,0.5,0.75}, we observe that memorization remains essentially unchanged, while generalization degrades progressively as the residual pathway is weakened, with the largest drop occurring at c=0 (full removal). This smooth trend shows that our findings are not an artifact of extreme ablation: even partial suppression of the residual connection does not stop memorization and consistently harms test accuracy. Together, these results reinforce our main conclusion that residual connections transmit generalization signals but do not propagate memorization.
> >
> >
> >
> > >**Q2:** Since the parameter count does not change with or without residual connections, might the memorization metric be essentially determined by model size? Note that from a training standpoint, if residual connections are entirely absent, increasing depth can cause vanishing gradients, so the difficulty of memorizing the data may simply increase.
> > >
> > >**Our Response:**
> > >
> > >Thank you for your discussion. Regarding the reviewer’s concern about parameter count: we would like to draw the reviewer's attention to our results in Fig. 1 comparing models of different sizes (GPT-2 Small vs. GPT-2 Medium), and in both cases memorization remains consistently unchanged even after residual connections are removed. This shows that the phenomenon is not tied to model size and that residual connections do not propagate memorization.
> >
> >
> >
> > >**Q3:** The current experiments use only 1% noise, which is a restrictive setting. Experiments with other noise levels, such as 5% or 10%, are natural to consider, so what is the reason for using only 1%?
> > >
> > >**Our Response:**
> > >
> > >We take into account the reviewer’s suggestions and carry out our analysis even on higher label noise ratios of 5%, 10%, and 20% and present the results in **Section 4.3**.  Clearly, from the results, we can see that our main claims remain valid even when the noise ratios are increased, i.e., (1) residual connections only propagate generalization but not memorization, and (2) early layers residual connections are the most impactful.
> >
> > **We hope our response resolves your concerns and questions, and helps you consider raising your score. If there is anything that we can further clarify, please let us know. We appreciate your time and effort again.**

---

> > > ### Comment · Reviewer_o1jb · 2025-11-22
> > >
> > > I appreciate the authors’ reply, and I see that the findings are interesting. However, I am still a bit confused about some of the paper’s main claims.
> > >
> > > > memorization gradient norms are smaller than learning gradient norms across all layers. Therefore, providing a direct indication of why residual connections do not propagate memorization
> > >
> > > If the gradient norm were zero, one could say that residual connections “do not propagate memorization.” Could you elaborate on how we can conclude one of the main contributions of the paper from the magnitude relationship between the memorization and learning gradient norms?
> > >
> > > Thank you also for providing the results for other levels of label noise. Could you additionally report the numerical memorization rates, rather than only the bar plot? Is the memorization still 100% in these settings, as claimed in the main paper?

---

> > > > ### Author Response · Authors · 2025-11-22
> > > >
> > > > **We thank the reviewer very much for promptly getting back to us. Below we address their follow-up questions:**
> > > >
> > > > ---
> > > >
> > > > **Q1 Response:**
> > > >
> > > > As shown in Figs. 5a and 5b, the memorization gradient norms are significantly smaller than the learning gradient norms across all layers. This essentially implies that the residual connections have almost negligible contribution to memorization in comparison to generalization, because the learning gradient norms are very high. Hence, it explains that removing the residual connections does not influence memorization but only generalization. We hope this answers the reviewer’s question.
> > > >
> > > > ---
> > > >
> > > > **Q2 Response:**
> > > >
> > > > Across all noise ratios, the memorization score after removing the residual connections remains extremely close to the score before the removal, supporting the claim that residual connections do not propagate memorization in transformers. Please find below the tables of numerical memorization score for the newly added plots for higher label noise ratios: (1) Smol-LM - 5%, 10%, 20%, and (2) DeiT - 5%, 10, 20%, presented in Section 4.3 and Appendix E.2.
> > > >
> > > > **Results for Smol-LM:**
> > > >
> > > > |**Label Noise Ratio**|**No Removal**|**Layer 1**|**Layer 3**|**Layer 4**|**Layer 6**|**Layer 8**|**Layer 12**|**Layer 16**|**Layer 20**|**Layer 24**|**Layer 28**|**Layer 29**|Layer 30|
> > > > |-----------------|----------|-------|-------|-------|-------|-------|--------|--------|--------|--------|--------|--------|--------|
> > > > |5%               |`100`       |100    |100    |100    |100    |100    |100     |100     |100     |100     |100     |100     |100     |
> > > > |10%              |`100`       |99.98  |100    |100    |100    |100    |100     |100     |100     |100     |100     |100     |100     |
> > > > |20%              |`100`       |99.59  |100    |100    |100    |100    |100     |100     |100     |100     |100     |100     |100     |
> > > >
> > > >
> > > >
> > > > **Results for DeiT:**
> > > >
> > > > |**Label Noise Ratio**|**No Removal**|**Layer 1**|**Layer 2**|**Layer 3**|**Layer 4**|**Layer 5**|**Layer 6**|**Layer 7**|**Layer 8**|**Layer 9**|**Layer 10**|**Layer 11**|**Layer 12**|
> > > > |-----------------|----------|-------|-------|-------|-------|-------|-------|-------|-------|-------|--------|--------|--------|
> > > > |5%               |`99.6`      |99.34  |99.34  |98.81  |99.08  |99.21  |99.34  |99.47  |99.08  |99.34  |99.87   |99.87   |99.6    |
> > > > |10%              |`99.34`     |99.27  |98.95  |99.14  |98.95  |99.47  |99.27  |99.34  |99.14  |99.21  |99.21   |99.14   |99.27   |
> > > > |20%              |`99.21`    |98.98  |98.65  |98.91  |98.71  |98.71  |98.95  |99.04  |99.01  |99.04  |98.98   |99.14   |99.18   |
> > > >
> > > >
> > > > ---
> > > >
> > > > \
> > > > **We hope our response resolves your questions. If there is anything that we can further clarify, please let us know. We appreciate your time and effort again.**

---

> > > > > ### Comment · Reviewer_o1jb · 2025-11-25
> > > > >
> > > > > Thank you for the answers. I feel that the main contribution is somewhat overstated; for example, the statements that residual connections **do not** propagate memorization and **do not** contribute to memorization seem a bit too strong. The theoretical analysis only establishes a magnitude relationship between the gradient norms, whereas the empirical analysis illustrates the implications of these findings under certain settings. In fact, the additional results appear not to be fully consistent with the authors' claim that 100% memorization holds. In the initial submission, the experiments seemed to be designed to align closely with the main claims, which is why the other reviewers asked to see results for higher levels of label noise beyond 1%.

---

> > > > > > ### Author Response · Authors · 2025-11-25
> > > > > >
> > > > > > Thank you for your response. We stated “(nearly identical) 100%” although on average it is  **99.982%** for all the numbers that we reported in the paper and responses. (Please note that it should be read from the difference between **No Removal vs. After Removal**.) If you strongly suggest, we will correct the number everywhere, or we will add “nearly” or “approximately” where we missed. Please advise us if it would sound better if we state, “residual connections propagate **0.018%** of memorization”. Otherwise, we would greatly appreciate it if the reviewer could suggest a more suitable statement based on the results that the reviewer had in mind.

---

### Official Review · Reviewer_iP5V · 2025-11-01

**Soundness:** 2
**Presentation:** 3
**Contribution:** 2
**Rating:** 4
**Confidence:** 4

**Summary:**

The paper asks whether residual (skip) connections in transformers propagate memorization or merely aid learning/generalization. The authors inject 1% label noise into several benchmarks, train models to 100% training accuracy, and then study (i) accuracy on clean test sets (a proxy for generalization) and (ii) accuracy on the mislabeled training points (their memorization metric). Across 8 transformer variants spanning vision and text (e.g., ViT‑Base, TinyViT, BEiT, DeiT, GPT‑2 Small/Medium, Qwen2‑0.5B, Smol‑LM) and 7 datasets, they report that removing residual connections hurts test accuracy but leaves "memorization" essentially unchanged. Layerwise ablations further suggest early residuals matter most for generalization. They support these findings with analyses of gradient norms w.r.t. residual-stream inputs, an output‑margin study, and two theorems giving upper bounds that attribute the memorization/learning gap primarily to the MSE betweenn y_hat and y and to multiplicative depth factor.
Overall, this is an interesting paper but I think requires some revisions before being ready for publication

**Strengths:**

Measuring gradients w.r.t. residual‑stream inputs and relating them to output margins and accuracy (Figs. 3, 6, 7) gives a coherent picture of where learning signal flows

The cross‑modal, multi‑model layerwise study is thorough and consistent. The effect shows that early residuals are far more consequential for test accuracy than later ones, while the chosen memorization metric barely moves

**Weaknesses:**

The memorization story is, in its current form, undermined by a narrow metric and by an experimental setup that can make "100% memorization regardless of residuals" nearly tautological. The theoretical analysis is intuitive but rests on sigma-based bounds whose applicability to trained, normalized transformers is not fully substantiated.


Measuring only label‑noise fitting on small‑/mid‑scale classification fine‑tunes misses prominent memorization phenomena in transformers (e.g., verbatim recall/PII leakage in generative LMs). Claims like “residuals do not relay memorization” are too broad given the limited metric and tasks

**Questions:**

N/A

---

> ### Author Response · Authors · 2025-11-18
> **Response to reviewer iP5V (1/2)**
>
> **We thank the reviewer for their comments and feedback on our paper. Below, we address their questions in detail, while providing the requested additional experiments.**
>
> > **Weaknesses:**
> >
> > **W1:** The memorization story is, in its current form, undermined by a narrow metric and by an experimental setup that can make "100% memorization regardless of residuals" nearly tautological. The theoretical analysis is intuitive but rests on sigma-based bounds whose applicability to trained, normalized transformers is not fully substantiated.
> >
> > **Our Response:**
> > Thank you for your comment. We believe the reviewer’s concern stems from a misunderstanding of our motivation and experimental setup. Therefore, please let us try to re-iterate and make clearer our core contributions and motivation.
>
> > As explained in the Introduction, our scope of interest is dedicated to residual connections. We were genuinely wondering whether residual connections can propagate either or both memorization and generalization, from one layer to the other, even though they do not consist of any model parameters. Furthermore, there is no existing work that studied their impact on memorization, making our work cleanly **novel**. We genuinely believe someone in the research community must answer this question, as no existing work has done it. As models grow more complex, understanding which components carry and do not carry memorized information is crucial for model interpretability and improved model design.
>
> > To evaluate memorization, we adopt the standard memorization metric used in prior work [1,2,3], ensuring that our experimental setup is consistent with established methodologies. Importantly, we would like to highlight that our finding that the model memorizes 100% of the noisy samples even after removing residual connections is not an obvious tautological outcome. This is primarily because, as stated in our motivation, no prior work has articulated whether residual connections can propagate either or both memorization and generalization. Hence, understanding whether their removal affects memorization is a meaningful and nontrivial question.
>
> > To understand why residuals do not propagate memorization, we conduct a gradient analysis. Specifically, we measure memorization and learning gradient norms for every layer. As shown in Fig. 3, memorization gradient norms are consistently far smaller than learning gradient norms, indicating that residual pathways primarily support generalization rather than memorization.
>
> > Our theoretical analysis formalizes the behavior. **Theorem 1** provides an upper bound on the residual gradient norm in terms of prediction error, residual statistics, and model parameter statistics. This theorem utilizes well-defined sigma-based upper bound analysis provided in [4] (please refer to **Eq. 17 from lines 835–839** in Appendix A) to prove Theorem 1 in Appendix A. The bound explains our empirical observations: memorization gradients remain small because they are controlled largely by prediction error, not by residual connection properties or parameter statistics, where prediction error for memorization samples is much smaller than learning samples (shown in Fig. 5), and thereby making memorization gradient norms smaller than learning gradient norms. **Theorem 2** extends Theorem 1 to show why early residuals have larger upper bounds of the gradient norms than later ones, which supports the strong empirical influence of early residuals observed in Fig. 2.
>
> > To further investigate whether the statistics of residual connections affect the influence of early layers, we conducted an additional σ-analysis (standard deviation analysis) as shown in Fig. 4. We found that the standard deviation of the residual connections, σₓ, is significantly smaller in early layers compared to later layers (Fig. 4b and 4c). This smaller σₓ amplifies the influence of early layers beyond the multiplicative depth factor. This is because, according to Theorem 1, σₓ and the upper bound of the gradient norm are inversely related, which explains this amplification effect. To the best of our knowledge, this phenomenon has also not been shown in prior work.
>
> >Together, our empirical findings and theoretical results provide a coherent explanation:
> **(1) residual connections do not meaningfully propagate memorization signals,
> (2) early residuals disproportionately support generalization, and (3) gradient level analysis is supported by well-grounded theorems** Neither phenomenon has been addressed in prior literature, establishing the novelty of our work.
>
> We hope this clarification helps resolve the reviewer’s confusion regarding the core novel  contributions of our work.

---

> > ### Author Response · Authors · 2025-11-18
> > **Response to reviewer iP5V (2/2)**
> >
> > > **W2:** Measuring only label‑noise fitting on small‑/mid‑scale classification fine‑tunes misses prominent memorization phenomena in transformers (e.g., verbatim recall/PII leakage in generative LMs). Claims like “residuals do not relay memorization” are too broad given the limited metric and tasks
> > >
> > > **Our Response:**
> > We understand the reviewer’s point of view and hence provide analysis on generative tasks, while considering the “extractable memorization” setting proposed in [5]. Please refer to Section 4.4 in the revised PDF for a thorough discussion. In summary, our main claims - (1) residual connections do not propagate memorization and only transfer generalization, (2) early residuals are the most impactful, hold true in generative tasks.
> >
> >
> > **We hope our responses satisfactorily address the reviewer’s concerns and help you to consider increasing your score. If any questions remain or further clarification is needed, we would be happy to provide additional details.**
> >
> > [1] Maini, Pratyush, et al. "Can neural network memorization be localized?." ICML 2023
> >
> > [2] Arpit, Devansh, et al. "A closer look at memorization in deep networks.", ICML 2017
> >
> > [3] Feldman, Vitaly, and Chiyuan Zhang. "What neural networks memorize and why: Discovering the long tail via influence estimation.", NeurIPS 2020
> >
> > [4] Takase, Sho, et al. "Spike no more: Stabilizing the pre-training of large language models.", COLM 2025
> >
> > [5] Carlini, Nicholas, et al. "Quantifying memorization across neural language models.", ICLR 2022

---

> ### Author Response · Authors · 2025-11-24
> **Gentle reminder to review the rebuttal**
>
> We would greatly appreciate it if the reviewer could review our rebuttal and let us know if our responses address their concerns. We would be glad to provide additional clarification or engage in further discussion as needed.

---

> > ### Comment · Reviewer_iP5V · 2025-11-24
> >
> > I appreciate the authors efforts and revisions. I think my problem continues to be the conclusions that we draw from this work: residual connections are important for learning good task performance. We of course know that residuals are important for learning, and thus I do not think the finding that "residuals propagate generalization" but not memorization is a little too broad/clear from prior work (the resnet paper). It is my understanding that the paper retrains with and without residual connections. Training without residual connections means that the model can still learn something (likely just overfitting to the training data) without actually generalizing well. It would be more interesting if the residual connections were removed from a trained model this effect was present (i.e., didn't totally destroy the model's performance on either setting).

---

> > > ### Author Response · Authors · 2025-11-24
> > >
> > > **We thank the reviewer very much for promptly getting back to us. Below we address their follow-up questions:**
> > >
> > > Thank you for your follow-up. Yes, residual connections are utilized for stable gradient flow as discussed in the original ResNet paper. However, that does not necessarily mean residual connections convey no memorization. Prior studies [1,2,3] support this by showing that residual connections can have harmful impacts. This hints that the “information” transferred by residual connections may also contain negative, harmful signals. Since memorization hurts generalization and hence is harmful, based on prior work, it is fair to ask whether “residual connections can propagate memorization or not”. Thereby, making the contributions of our work novel.
> > >
> > > Regarding the second part of your question, although we are not sure if we understood your questions correctly, our best understanding is that the reviewer is asking what happens if residual connections are removed after the model has already been trained, rather than disabling them during training. To examine this, we conducted an additional analysis using a ViT-Base model trained on CIFAR-10. After training, we removed the residual connections one layer at a time and measured the resulting performance.
> > > The results (summarized in the table below) show that removing residual connections post-training severely destroys model’s performance, typically down to near-random performance (10-11%) when early-layer residuals are removed.
> > >
> > >
> > > | | No Residual Removal  | L1    | L2    | L3    | L4    | L5    | L6    | L7    | L8    | L9    | L10   | L11   | L12   |
> > > |------------------|-------|-------|-------|-------|-------|-------|-------|-------|-------|-------|-------|-------|-------|
> > > | **Test Accuracy (%)**| `92.62` | 10.14 | 11.29 | 10.82 | 10.99 | 10.41 | 10.57 | 15.66 | 27.01 | 29.28 | 43.56 | 79.05 | 91.40 |
> > >
> > >
> > > Please let us know if this aligns with the reviewer's intended meaning. We would be happy to provide any further clarifications as needed.
> > >
> > > ---
> > >
> > >
> > > [1] Wu, Dongxian, et al. "Skip connections matter: On the transferability of adversarial examples generated with resnets.", ICLR 2020
> > >
> > > [2] Hao, Koh Jun, et al. "On the Vulnerability of Skip Connections to Model Inversion Attacks.", ECCV 2024
> > >
> > > [3] Kelesis, Dimitrios, Dimitris Fotakis, and Georgios Paliouras. "Analyzing the effect of residual connections to oversmoothing in graph neural networks." Machine Learning 2025

---

### Author Response · Authors · 2025-11-18
**Global Response: Additional Experiments and Results added in the revised PDF**

We thank all reviewers for their time, effort, and thoughtful feedback. We carefully addressed every comment and conducted several additional experiments requested during the review process. The revised PDF now includes the following new results and analyses (marked in **blue** throughout the paper) based on the reviewers’ suggestions:

1. **Consistent Results for Higher Label Noise Ratios:** Evaluation under higher label noise ratios - 5%, 10%, and 20%, assesses robustness of our paper’s main **novel** findings. We discuss this analysis in **Section 4.3 and Appendix E.2**.

2. **Consistent Results for Generative Tasks:** Extension of our analysis to generative tasks, where we observe consistent findings that residual connections do not propagate memorization but only relay generalization, along with the impactful nature of early residual connections. We provide a detailed discussion regarding this in **Section 4.4 and Appendix E.3**.

3. **Output Margin and Gradient Norms analysis across epochs:** Epoch-wise tracking to provide deeper insight into how output margin and gradient norms gradually evolve across the course of training, with early residuals exhibiting higher gradient norms than middle and later ones throughout the course of training. The results corresponding to these are provided in **Appendix E.10**.

4. **Scaling residual connections:** Analysing how gradual removal of residual connections by multiplying them by a scaling factor influences memorization and generalization. Under this setting, it is again confirmed that residual connections do not propagate memorization, with a detailed discussion provided in **Appendix E.4**.

**We hope these additional experiments answer and resolve all of the reviewers’ questions regarding our paper. If there are any further unresolved questions, please let us know. We will be happy to answer all of them.**

---

### Author Response · Authors · 2025-12-01
**Global Response: Core Contributions of our Paper and Rebuttal Summary for the new AC**

Below we re-iterate the **motivation** and core **novel** contributions of our paper, along with a summary of the rebuttal for the new AC.

---

**1. Motivation and Core Contributions of our paper:** Residual connections propagate signals, but those signals may encode either or both memorization and generalization. Our paper addresses the novel and previously unexplored question: ***Do residual connections propagate memorization in transformers?*** We genuinely believe someone in the research community must answer this question, as no existing work has done it. Hence, through our paper we answer this question, supported by thorough empirical findings and theoretical results:
**(i) residual connections do not transmit memorization signals but rather only generalization; (ii) early residuals play an outsized role in enabling generalization; and**  **(iii) these observations are supported by gradient-level analyses and accompanying theorems.**

---

**2. Consistent Results for Generative Tasks:** As per the suggestion of reviewer ``iP5V``, we extended our analysis on **generative tasks**, where we observe consistent findings that residual connections do not propagate memorization but only relay generalization, along with the impactful nature of early residual connections. We provide a detailed discussion regarding this in Section 4.4 and Appendix E.3.

---

**3. Consistent Results for Higher Label Noise Ratios:** Based on the suggestions from reviewers ``o1jb``, ``Lgg1``, and ``Q3pD``, we extended our analysis for higher label noise ratios. We observe that even for various higher label ratios, our paper’s main findings remain consistent.  We discuss this analysis in Section 4.3 and Appendix E.2.

---

Overall, we would like to thank the AC for their time and effort in reviewing our paper.

---

### Meta-Review · Area_Chair_9JDN · 2025-12-26

**Summary:**

The core claim of the paper is that residual connections do not contribute to memorization, evidenced by the fact that retraining the same architecture without residual connections still enables the model to overfit a dataset with added uniform label noise.

The experimental scope is extensive, particularly in the revised version. Experiments across multiple datasets and models show consistent results. During the rebuttal phase, the authors addressed feedback from Reviewer iP5V by adding experiments with larger label noise and generative tasks.

A major weakness of the paper is the difficulty in verifying or falsifying the central hypothesis. Specifically, removing residual connections results in a fundamentally different model that may require different hyperparameters or alternative interventions (e.g., additional normalization common in pre-residual architectures) to function optimally. Therefore, it is unclear if retraining with the same hyperparameters is a sufficient test of the claim. This concern was reflected in comments by Reviewers o1jb and iP5V, who suggested alternative ways of establishing the claim. Most importantly, iP5V stated, which I agree with, that the result of memorizing dataset is "tautological" in the sense that any overparametrized model can overfit data.

Reviewers also disputed the conclusions drawn from the gradient norm analysis. While gradient norms are related to learning speed (via the first-order expansion of the loss function), this relationship should not be presented as a direct proof that residuals "do not relay" memorization.

Taken together—the difficulty of verifying the hypothesis and the presentation of indirect arguments as direct proofs—the paper appears to overclaim when interpreting its results. The fact that an architecture without residual connections can still overfit does not necessarily imply that residual connections play no role in memorization; it may simply mean the remaining parameterized layers are sufficient to achieve overfitting. As noted in prior research, networks without residual connections are known to be capable of overfitting uniform label noise. A more robust way to disentangle this impact might be to test the limits of memorization on significantly harder datasets or under extreme noise levels.

Three Reviewers voted for rejecting the paper while one voted for accepting it. Reviewers have not adjusted their score during the rebuttal phase.

Overall, while the paper presents interesting experiments, the interpretation of the results requires further clarification and more nuanced claims before it is ready for acceptance.

**Reviewer Concerns:**

The key concern was the formulation of the main claim and how it was tested. Reviewers o1jb and Q3pD argued that the findings are intuitive and unsurprising. I do not agree with this comment but I do think that it points to the claim being unclear. Reviewer iP5V criticized the setup of retraining models without residuals, arguing this creates a fundamentally different model where "100% memorization" is a tautological result of overparameterized training. This is at the core of the issue of the paper and Authors have not fully rebutted this.

**Reviewer Scores:**

Reviewers would likely not change significantly their scores because the core issue that the core claim is too simplistic/broad.

---

### Decision · Program_Chairs · 2026-01-26

Reject